# Emerging heterogeneous compartments by viruses in single bacterial cells

Jimmy T. Trinh[1,2], Qiuyan Shao[1,2], Jingwen Guan[1,2,3] & Lanying Zeng ![ORCID] [1,2,3 ✉]

Spatial organization of biological processes allows for variability in molecular outcomes and coordinated development. Here, we investigate how organization underpins phage lambda development and decision-making by characterizing viral components and processes in subcellular space. We use live-cell and in situ fluorescence imaging at the single-molecule level to examine lambda DNA replication, transcription, virion assembly, and resource recruitment in single-cell infections, uniting key processes of the infection cycle into a coherent model of phage development encompassing space and time. We find that different viral DNAs establish separate subcellular compartments within cells, which sustains heterogeneous viral development in single cells. These individual phage compartments are physically separated by the *E. coli* nucleoid. Our results provide mechanistic details describing how separate viruses develop heterogeneously to resemble single-cell phenotypes.

[1] Department of Biochemistry and Biophysics, Texas A&M University, College Station, TX 77843, USA. [2] Center for Phage Technology, Texas A&M University, College Station, TX 77843, USA. [3] Molecular and Environmental Plant Science, Texas A&M University, College Station, TX 77843, USA. ✉email: lzeng@tamu.edu

Organization is a fundamental part of life. For complex organisms, the spatial development of body parts is controlled for proper function[1]. In the cells comprising these organisms, separate organelles are organized by membranes[2]. Bacterial cells utilize proteins to localize processes in lieu of intracellular membranes[3]. Furthermore, the physical inhomogeneity of cytoplasmic DNAs, RNAs, and proteins favor the segregation of components for different processes[4–6]. Viruses are even simpler than cellular life, but have also been reported to organize their development within cytoplasmic inclusions or proteinaceous compartments[7,8].

Bacteriophages, bacterial viruses, are among the simplest biological systems and serve as models for advanced cellular processes. High-resolution fluorescence microscopy and mathematical modeling have been used to examine the phage lambda lysis–lysogeny paradigm for cellular decision-making to uncover surprising phenomena[9–11]. By labeling single virus particles, we characterized that separate votes for decisions by co-infecting lambda phages determined cell fates[12]. We next provided distinct voices for separate phages to expound upon our voting model by incorporating different lytic and lysogenic reporters into different phages, finding that phage DNAs compete and cooperate as subcellular individuals[13]. It is curious how identical viral DNA molecules can commit to divergent trajectories while inhabiting a single cellular environment. While stochasticity is understood to definitely influence development, where noisy elements of gene expression might partially explain differential subcellular behaviors[14], it is also evident that properly targeted, high-resolution studies can elicit molecular mechanisms beyond the intrinsic stochasticity of cellular biochemistry[15–17]. We hypothesize that subcellular organization allows different phages to develop as individuals in a single cell and expect to detect the underlying subcellular heterogeneity by specifically investigating the spatial distribution of viral/host biomolecules. Here, we show that cascading events during viral transcription, DNA replication, and gene expression combine to establish an organized subcellular unit of phage development. This organization permits multiple individual viruses in the same cell to develop via different pathways in separate areas of that cell.

## Results

### Live-cell fluorescent reporters of phage development.
To work toward a unified model of individual lambda development in cellular space, we used live-cell time-lapse microscopy targeting the initial infecting phage DNAs, host cell's replication resources, replicated phage DNAs, and phage decisions (Fig. 1a). To visualize the initial infecting phage DNA, we modified cells to be $dam^-$ and carry a $seqA-mKO2$ (mKO2 signal defined as yellow) translational fusion[18]. This system has been validated previously to correctly label single DNA molecules from infecting phages[11,13,18]. Upon ejection of methylated phage DNA into the host, SeqA-mKO2 binds exclusively to the ejected phage DNA and any DNA copies retaining the methylated parent strands with single-DNA labeling sensitivity, but not subsequent replicated DNA copies (SeqA system, Fig. 1b)[11,18]. Since the SeqA system cannot target all replicated phage DNAs, we recombineered an array of $tetO$ sequences into the phage genome. With the host cell harboring a TetR-mCherry plasmid (mCherry signal defined as red), all phage genomes are bound and labeled at $tetO$ sites by TetR (Tet system, Fig. 1d) (Supplementary Discussion). The Tet-labeling scheme lacks single-DNA sensitivity under our experimental conditions, so we used both Tet and SeqA systems to target phage DNA. As lambda depends on host factors for viral DNA replication, we translationally fused the *Escherichia coli* helicase[19], DnaB, with mTurquoise2 (mTurquoise2 signal defined

as blue) by replacing the native $dnaB$ gene with $dnaB-mTurquoise2$ on the *E. coli* chromosome (Fig. 1c; Supplementary Fig. 2f). DnaB is essential for phage/*E. coli* DNA replication and directly interacts with lambda P (analog of *E. coli* DnaC)[20]. The DnaB construct does not appear to impose major detriment on *E. coli* or phage growth (Supplementary Fig. 2g–j). Finally, we reported lambda lysis–lysogeny decision-making using previously developed systems[13]. Briefly, we modified phages with a *D-mNeongreen* (mNeongreen signal defined as green) translational fusion, reporting the lytic pathway because progeny phages are assembled with green gpD, and a $cI-mKO2$ transcriptional fusion, reporting the lysogenic pathway because the $cI$ operon(s) are expressed during lysogeny (Fig. 1a). Accordingly, we developed a data analysis framework for these reporters to detail the spatial organization of subcellular events during infection (Supplementary Fig. 1). Notably, all presented images of individual cells unambiguously represent single cells, because early expression of the Kil protein by lambda inhibits cell division during infection[21].

### Organization of resources and replication by individual phages in single cells.
We expect that compartmentalization of biomolecules reflects the establishment and perpetuation of individualistic development by lambda. Uninfected cells (LZ1557) displayed diffuse blue, yellow, and red fluorescence, indicating that DnaB, SeqA, and TetR do not compartmentalize without phage infection (Supplementary Fig. 2a; Supplementary Movie 1). Functional DnaB is required for growing cells[19]; thus, DnaB localization under cell growth conditions represents a basal DnaB state. Importantly, active DnaB-mTurquoise2 does not localize as a focus under phage-free, normal growth conditions. When infecting these cells with our phages (λLZ1576), the fluorescence patterns change significantly from the phage-free state. We primarily focus on analyzing lytic cells because extensive DNA replication is required for successful lytic propagation and note that lysogenic development markedly differs from lytic behaviors (Supplementary Fig. 3b, c; Supplementary Movie 2; Supplementary Discussion). SeqA foci appear within the cells, typically early during infection (Supplementary Fig. 3a), indicating that phage DNA entered the cell (Fig. 1e). As phage DNA replication demands host resources, we tracked DnaB. DnaB foci typically formed after SeqA foci and were commonly colocalized with SeqA foci over time (Fig. 1e, f, i, j; Supplementary Fig. 4a, b). This indicates that phage DNA directly alters the natural behavior of DnaB by collecting essential resources to its own location. DnaB foci represent multiple DnaB complexes, specifically aggregated to single phage DNAs for extensive phage DNA replication. In lytic cells, blue foci remained over time (median: 90 min), suggesting that lambda persistently recruits resources for itself (Supplementary Fig. 3b).

Resource recruitment precedes phage DNA replication. Red fluorescence is initially spread throughout the cell, representing free TetR as background signal, not phage DNA signal (Supplementary Fig. 2c, e, at 40 min). TetR signal later rearranged into small clusters over time, signifying the production of additional phage DNAs nearby the specific locations of the single DNAs and resources (Fig. 1e–h; Supplementary Movie 3). These subcellular local maxima, or clusters, of TetR signal represent phage DNA. DnaB was pre-localized near the eventual red clusters and remained within the clusters as they expanded within the cell, suggesting that replicating phage DNAs predictably arise at the location where previous DNAs gathered resources, and then maintain this sequestration (Fig. 1k, l; Supplementary Figs. 3d and 4c, d). In lytic cells, gpD (green) signal increased throughout the cell over time (Fig. 1e–h). This signal is initially diffused throughout the cell, but later, green foci, corresponding

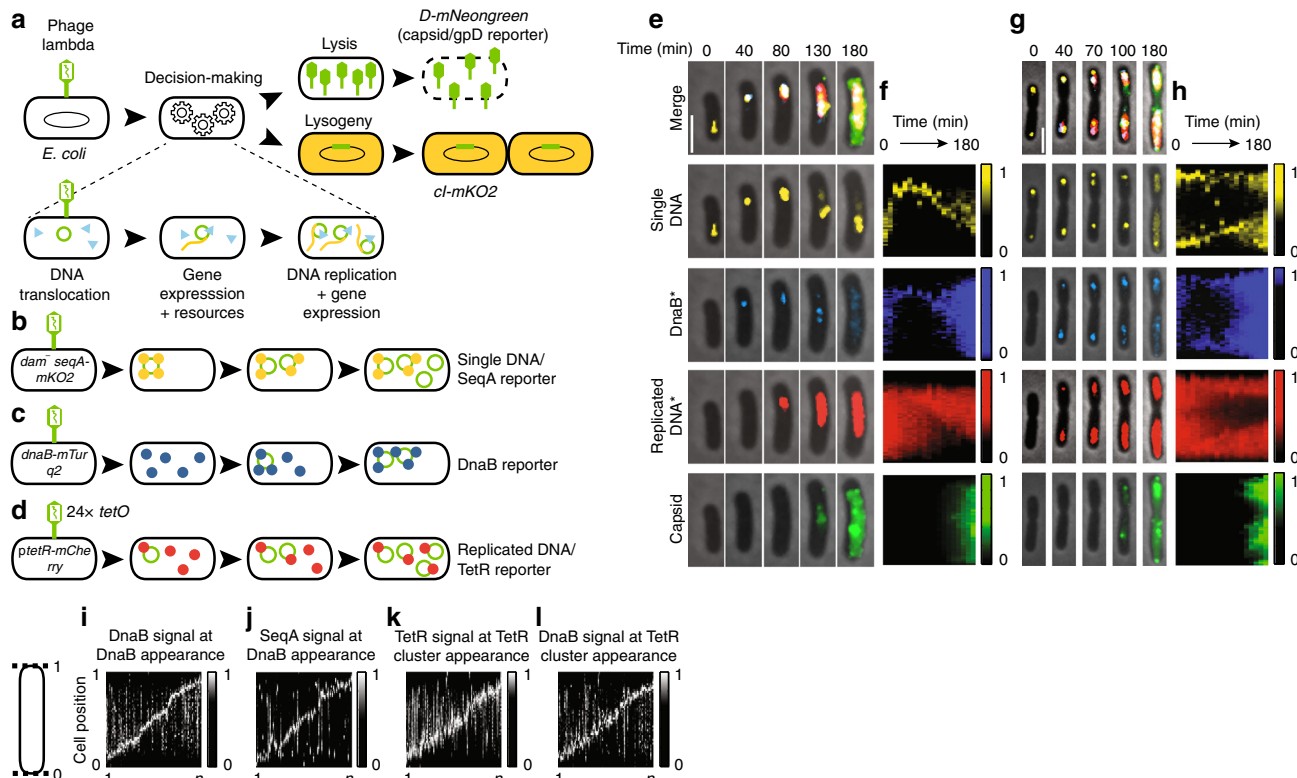

**Fig. 1 Phage DNAs organize developmental processes into subcellular locations during infection. a** Combination of phage processes results in decision-making. Lytic decisions reported by a *D-mNeongreen* translational fusion and lysogenic decisions reported by a *cI-mKO2* transcriptional fusion. **b** SeqA system detects single molecules of phage DNA. Methylated phages infect *dam⁻* cells. Phage DNA is bound by SeqA-mKO2 proteins. Only phage DNA retaining methylation is labeled. **c** DnaB is an essential DNA replication resource. DnaB-mTurquoise2 fusion protein reports localization of DnaB. **d** Tet system detects replicated phage DNAs. Phage DNA bearing *tetO* arrays is labeled by TetR-mCherry binding. **e** Representative infected cell with reporters described in **a**–**d** undergoes lytic development. Representative cells in **e** and **g** chosen from three independent infection experiments. * indicates contrast is adjusted for each time point for clarity. DnaB and replicated DNA fixed contrast images are shown in Supplementary Fig. 2b–e. All scale bars in this figure are 2 µm. **f** Kymograph of the cell in **e**. Explanations for data analysis in Supplementary Fig. 1 and Supplementary Discussion. Fluorescence is normalized to the population maximum. **g** Representative infected, lytic cell with two subcellular areas of development. **h** Kymograph of the cell in **g**. Fluorescence is normalized to the population maximum. **i** DnaB heat maps for lytic cells at their DnaB appearance time point are arranged by the position of DnaB. Cell to the left of **i** describes how location is represented for **i**–**l**. Fluorescence of each cell is normalized to its own peak brightness for **i**–**l**. *n* = 91 cells for **i**–**l**. **j** SeqA heat maps for lytic cells are arranged in the same order and time points as in **i** to compare SeqA and DnaB. **k** TetR heat maps for lytic cells at their TetR cluster appearance time point are arranged by the position of TetR. **l** DnaB heat maps for lytic cells are arranged in the same order and time points as in **k** to compare DnaB and TetR. Source data are provided as a Source Data file.

to phage capsids, preferentially formed in the red clusters (Supplementary Fig. 5a). This suggests that the locations of phage DNA determine where progeny phages are assembled, consistent with the characterized mechanisms of phage DNA packaging[22]. Altogether, the data indicate that a single phage DNA organizes its own subcellular phage factory, persistently maintaining its clones and resources proximal to itself. We designated this phage-derived, subcellular compartment as a "phactory" for future reference.

As the single phage DNA had the capacity to assemble its own compartment, we predicted that multiple, individual phage DNAs could form separate phactories. We identified cells with single phage DNAs in different locations (Fig. 1g, h; Supplementary Movie 4), where each DNA collected its own stockpile of DnaB. Separate red clusters appeared and grew at each DnaB location, and finally, green foci grew into clusters nearby the separated DNA clusters. Intriguingly, the phage-related biomolecules remain separated in space as different viral microenvironments, where the levels and identities of the phage DNA, RNA, and protein in the microenvironments may change and be exchanged while still maintaining spatial separation. These data suggest that each subcellular phage DNA can be its own entity and can

organize its own phactory within single cells. In conditions that specifically stimulate spatially segregated phage DNAs to develop, lysogenic induction (Fig. 2a; Supplementary Discussion), multiple phactories separately form and progress in single cells (Fig. 2b–d; Supplementary Figs. 5b and 6). Phactories have unequal DNA and lytic reporter levels over time, suggesting that heterogeneous development is sustained within individual phactories (Fig. 2e; Supplementary Figs. 7 and 8).

**Host nucleoids maintain separation of individually developing phages**. Our results indicate that phage DNAs in different phactories remain separated as the basis of their individuality, but do not indicate obvious barriers that segregate phage DNAs, obscuring the detailed mechanisms of their heterogeneous development. We observed that the growth of phage DNA clusters decreased as the cell filled with phage DNA, as if their size might be physically limited by something in the cell (Supplementary Fig. 9a, c; Supplementary Discussion). We hypothesized that replicating phage DNAs are physically separated from *E. coli* DNA, a presumed barrier shaping the phactories. We tested this hypothesis by labeling *E. coli attB* with a *lacO* array and LacI-EYFP construct, adapted from previous work[11]. We infected these

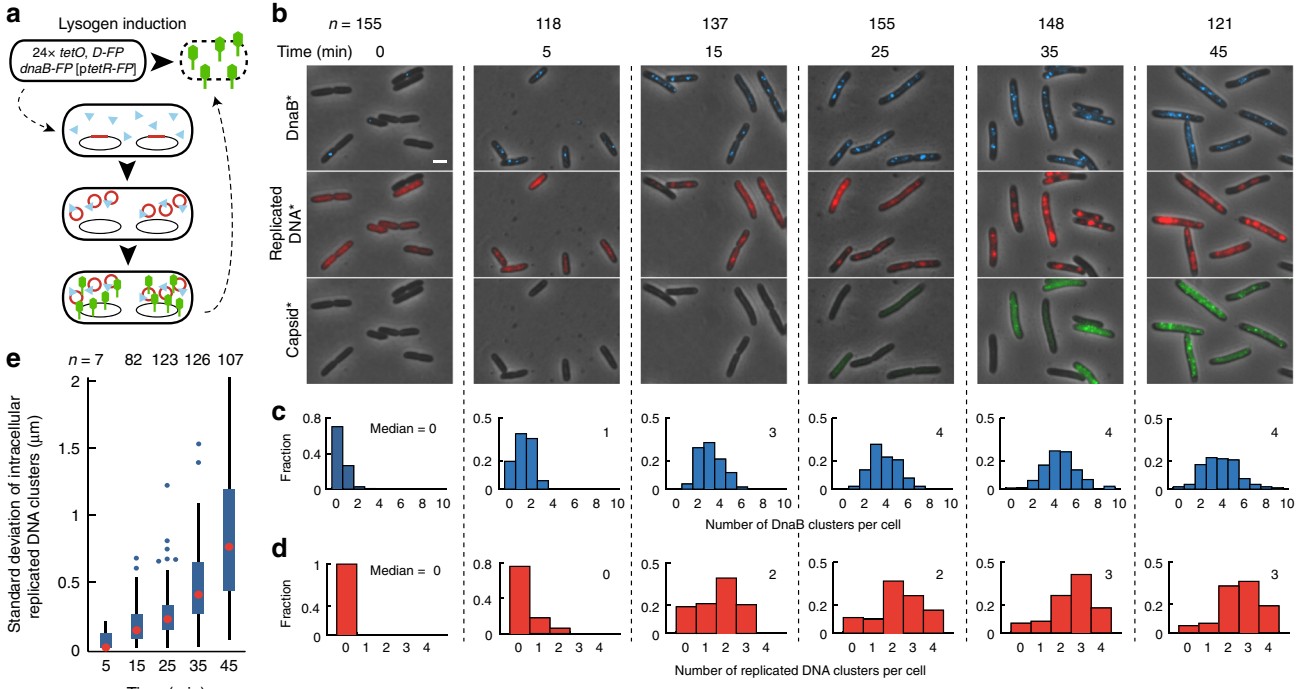

**Fig. 2 Organization of multiple heterogeneous subcellular areas during phage induction. a** Schematic for the spatial organization of phage development after induction. Lambda prophage is integrated into the bacterial chromosome, and different copies of the chromosome locate to different areas of the cell during growth. Induction of lysogens forces phages to develop in different areas of the cell. The prophage bears the gpD-mNeongreen lytic reporter and carries a *tetO* array. The cell harbors a TetR-mCherry plasmid and a DnaB-mTurquoise2 reporter. **b** Overlay images of lysogens after induction. At 0 min the cells were not yet induced (*indicates that the contrast is adjusted for each time point shown for clarity). All scale bars in this figure are 2 μm. **c**, **d** Intracellular areas of phage DNA and DnaB form. Histograms of the number of DnaB (**c**) and replicated DNA clusters (**d**) are shown for each time point. **e** Phage DNA replication varies intracellularly. For cells with more than one DNA cluster, the standard deviation of the size of the clusters is represented in boxplots for each time point, as a measure of intracellular phage DNA variability. The median is indicated by the dot at the center of the box, the box bounds the interquartile range of the data, the whiskers span the range of the data excluding the outliers, and the outliers are indicated as individual points. The approximate limit of our resolution is around 250 nm. Source data are provided as a Source Data file.

cells (LZ1643) with phages (λLZ1629) (Fig. 3a). Phage DNAs and capsids behave similarly to the above infections with λLZ1576 and LZ1557 (Supplementary Figs. 5c, 9b, d and 10). The terminus of lytic cells is cell lysis, so we tracked the location of *attB* in the final time points to determine how preceding lytic development affects *E. coli* DNA, finding that *attB* locations were biased near the poles (Fig. 3h). Conversely, the terminus of non-lytic development is cell division, and *attB* was localized between the mid-cell and quarter-cell in non-lytic cells, suggesting differential development (Fig. 3h; Supplementary Fig. 11a; Supplementary Movie 5). We subcategorized the lytic cells based on the interaction of phage DNA with *attB* (push, spread, and squeeze, Fig. 3b-g; Supplementary Fig. 11b-d; Supplementary Movies 6–8). In the largest class (push, 60%, 108 out of 179 lytic cells), we found that *attB* was pushed towards one side of the cell, away from a single expanding phage DNA cluster (Fig. 3b, c; Supplementary Movie 6). Within this population of cells, as phage DNA cluster sizes increased over time, *attB* moved closer to cell poles (Fig. 3i, j). Furthermore, after phage DNA clusters expand near *attB*, they do not move past it, suggesting that phage and bacterial DNA do not mix together (Supplementary Fig. 11e, f; Supplementary Discussion). These data agree with our previous observations regarding the polar movement of *attB* in lytic cells[11], provide a clear phage-active mechanism, where the spatial expansion of phage DNA due to replication explains the results, and corroborate our hypothesis that *E. coli* DNA helps determine phactory localization and segregation. Because the two phactories do not merge around the bacterial marker (Fig. 3f), it suggests

that bacterial DNA is a physical barrier which allows subcellular viruses to maintain different identities.

We next used alternate techniques to examine phage infection, utilizing bacteria/phages without genetically engineered reporters to minimize the likelihood that the genetic modifications influenced biological behaviors. We performed single-molecule fluorescence in situ hybridization (FISH) to characterize the spatial aspect of phage transcription and to examine phage DNA replication in fixed cells, to better characterize how different subcellular viruses develop[23–25].

To compare FISH with our live-cell techniques, we labeled phage λLZ613 DNA during infection using lambda DNA-specific probes. At early time points, phage DNA existed as small foci/clusters in cells (Fig. 4a). The localization of FISH foci resembles that of SeqA foci in live cells (Fig. 4p, q). At later time points, we found that DNA signal increased with time, and appeared as larger clusters, similar to our Tet reporter system (Fig. 4b). The amount of DNA per cell and within clusters varied (Fig. 4c, d; Supplementary Fig. 12a). To study bacterial DNA, we treated the cells with DAPI in FISH experiments. While DAPI stains DNA non-specifically, the size of *E. coli* DNA (4.6 Mbp) far outstrips that of lambda DNA (48.5 kbp); thus, DAPI will primarily stain bacterial DNA at early time points before extensive phage DNA replication, allowing phage DNA FISH signal and DAPI to be mutually exclusive (Supplementary Discussion). We observed that the spatial distribution of the DAPI signal in *E. coli* leaves distinct nucleoid-free zones (Fig. 4h, p; Supplementary Fig. 13b), and phage DNAs preferred these areas in early time points

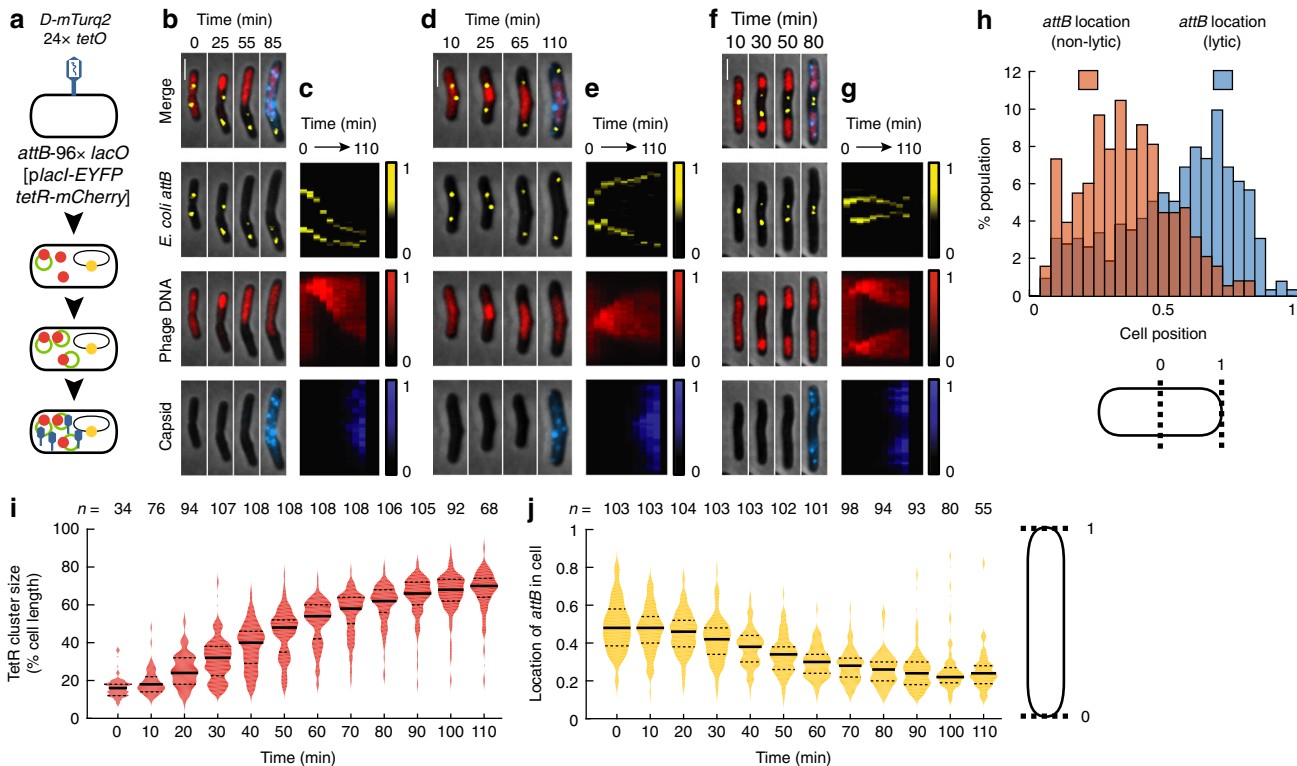

**Fig. 3 Bacterial DNA physically separates different intracellular phactories. a** Detection of phage and bacterial DNA locations. Phage carries the Tet reporter as in Fig. 1d and the *D-mTurquoise2* lytic reporter. Cell carries a *lacO* array at its *attB* locus and bears a plasmid expressing both TetR-mCherry and LacI-EYFP to label phage and bacterial DNA. **b, d, f** Representative lytic cells with different bacterial DNA interactions. Phage DNA will push (**b**, 108/179 cells), spread (**d**, 56/179 cells), or squeeze (**f**, 15/179 cells) bacterial DNA as phage DNA expands. Representative cells chosen from four independent infection experiments. All scale bars in this figure are 2 μm. **c, e, g** Kymographs corresponding to the cells in **b, d, f**. Fluorescence is normalized to the population maximum. **h** Spatial distribution of bacterial DNA in lytic and non-lytic cells differs. In non-lytic cells, the locations of *attB* for three time points prior to cell division (orange), and in lytic cells, the location of the *attB* marker for three time points prior to cell lysis (blue) represent the location preference of bacterial loci in different developmental paths. The cell below shows how locations are represented. *n* = 653 data points for lytic and 382 data points for non-lytic categories. **i, j** Expanse of phage DNA pushes bacterial DNA. The sizes of phage DNA in each lytic cell that pushes bacterial DNA are shown as violin plots (**i**). The maximum (nearest to mid-cell) location of *attB* of the cells in **i** are shown as violin plots (**j**). Cell on the right shows how locations are represented for **i, j**. Cells were oriented such that the TetR clusters were all aligned toward one direction. In the violin plots, the solid line represents the median, and the dashed lines mark the bounds of the interquartile range of the data. Source data are provided as a Source Data file.

(Fig. 4g, h, j, p; Supplementary Fig. 13a, b), supporting earlier observations of negative bacterial and phage DNA spatial correlation (Fig. 3). Phage DNA and DAPI signals overlap in later time points, after presumed phage DNA replication (Fig. 4j; Supplementary Figs. 12b–e and 13a, b, g). Furthermore, we generated probes against *E. coli attB*, and found that *attB* generally localizes with DAPI (Supplementary Fig. 13d–f), but avoids phage DNA locations (Fig. 4g, i, k; Supplementary Fig. 13a, c, h), congruent with our live-cell data (Supplementary Discussion). These results suggest that the organization of heterogeneous, separated phage DNA units exists and is reported faithfully by both live-cell and fixed-cell methods.

**Phages maintain localized transcription.** Given the organization of segregated phage DNAs, our next step was to investigate the organization of phage mRNAs because gene expression is a key component of phage development and decision-making. We studied phage transcription with RNA FISH[24], where we first targeted pR, an early transcriptional unit comprising genes for decision-making and phage DNA replication[26]. At early time points, pR transcripts existed as small clusters (Fig. 4e). We found that pR localizes to areas with lower DAPI signal (Fig. 4l–o; Supplementary Fig. 14a–d), near poles and/or mid-cells, regions well-characterized to typically be nucleoid-free[27,28], suggesting

that phage mRNAs, like phage DNAs, reside away from nucleoids. Remarkably, even at later time points, pR retains subcellular localization as opposed to diffusing evenly throughout the cell (Supplementary Fig. 14a). Notably, DAPI signals increase in pR locations at later time points following presumed phage DNA replication. From the convergence of our experimental data, we concluded that the locations of pR transcripts in FISH also represent the locations of phage DNA. These results suggest that intracellular phactories possess their own gene expression profiles in single cells. This surprising degree of mRNA spatial organization for lambda may be promoted by the physical attachment of mRNAs to phage DNAs during transcript elongation[29], and whole-cell diffusion may be discouraged by the relatively short lifetimes of mRNAs in *E. coli*[30], combined with similarly localized ribosomes translating the transcripts[27,31]. Notably, phage mRNA localization remains even after transcription is finished and further transcription is blocked with rifampicin treatment[32], demonstrating that phage RNAs are continuously localized with phage DNAs in the phactory (Supplementary Fig. 15).

**Separate decisions made in different locations of the same cell.** Downstream of phage DNA replication and early gene expression is decision-making, and we predicted that the organization of upstream processes would impact the execution of specific cell-

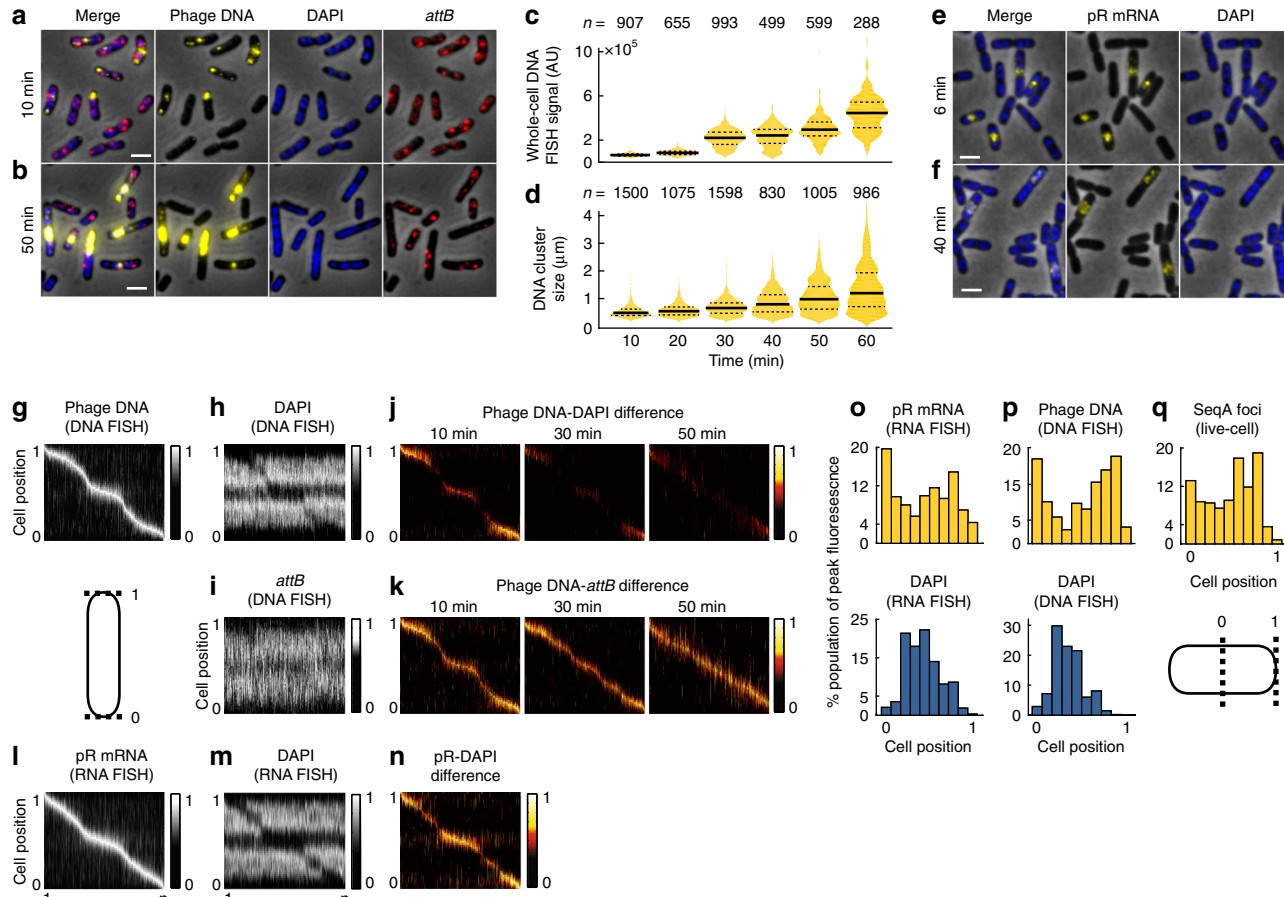

**Fig. 4 Phage transcription and DNA replication are collectively organized within nucleoid-free regions. a, b** DNA FISH labels phage and bacterial DNA. Representative images of DNA FISH experiments targeting phage DNA and *attB* at 10 min (**a**) and 50 min (**b**) post-infection with cells stained with DAPI. Representative cells chosen from six independent infection experiments. All scale bars in this figure are 2 μm. **c, d** Violin plots of whole-cell phage DNA signal (**c**) and sizes of phage DNA clusters (**d**). In the violin plots, the solid line represents the median, and the dashed lines mark the bounds of the interquartile range of the data. **e, f** RNA FISH labels phage pR transcript at 6 min (**e**) and 40 min (**f**) post-infection. Cells stained with DAPI. Representative cells chosen from six independent infection experiments. **g–i** Phage DNA (**g**), DAPI (**h**), and *E. coli attB* (**i**) heat maps arranged by the location of peak brightness of phage DNA at 10 min post-infection. Cell below (**g**) shows how locations are represented for **g–n**. Fluorescence of each cell is normalized to its own peak brightness for **g–i** and **l**, **m**. n = 907 cells for **g–i**. **j**, **k** Phage DNA prefers nucleoid-free locations and avoids *attB*. **j** Difference maps (see Supplementary Discussion), calculated as **g**, **h**, show contrast in locations of phage DNA and DAPI. Negative values are set to 0. **k** Difference maps as in **j**, except showing differences between phage DNA and *attB*. n for **j** and **k** = n of **c** for each time point. **l–n** Phage mRNA prefers nucleoid-free locations. pR RNA FISH (**l**) and DAPI (**m**) heat maps arranged by the location of peak brightness of pR at 15 min post-infection. **n** Difference map of **l**, **m**, similar to **j–k**. n = 2035 for **l–n**. **o–q** Phage DNA and mRNA colocalize away from *E. coli* DNA. **o** Histograms of peak location of pR and DAPI from RNA FISH in **l**, **m**. **p** Histograms of peak location of phage DNA and DAPI from DNA FISH (**g**, **h**). **q** Histogram of peak location of SeqA signal from experiments in Fig. 1 (lytic cells, combined first three time points). Cell below (**q**) shows how locations are represented for **o–q**. Source data are provided as a Source Data file.

fate expression programs[26]. The pR′ transcript encodes the lysis and morphogenesis proteins during lytic development; the pRE/pRM (referred to as pRE) transcripts encode CI to establish and maintain lysogeny. Therefore, we targeted the pR′ and pRE transcripts with FISH to characterize subcellular decision-making in subcellular space (Fig. 5a, b).

We performed a set of FISH experiments at 15 min after infection because decision-making is expected to occur around this timeframe[24]. In support of our model, pR levels vary at different locations of the same cell, more so than pR varies between different cells (Supplementary Fig. 14g). These data reiterate that pR expression is localized and suggest that the variation of localized pR mRNA levels may lead to different localized decisions. The lysogenic decision will eventually repress transcription from pR due to the action of CI[26]. Accordingly, we found that cellular pRE levels negatively correlate with cellular pR levels (Fig. 5c). Furthermore, focusing on the spatial aspect of

transcription, we determined that the cellular coordinates of the brightest pRE signals are offset from the brightest pR signals (Supplementary Fig. 14e, h). These data corroborate earlier hypotheses that pRE/pR head-on transcription might commit the lambda genetic circuit towards lysogeny when pRE dominates[33]. The data suggest that lysogenic decisions may be locally initiated by individual DNAs, where pRE activation halts pR transcription in a single phage genome, and that these localized actions precede cell-wide repression by CI. As for the pR′ transcripts, our FISH data show that the brightest pR′ and pR signals coincide within the cell. Altogether, the data suggest that decisions are enacted by separate phage DNAs in specific subcellular areas (Supplementary Fig. 14f, i).

## Discussion
Our model for lambda decision-making involves voting for decisions by co-infecting phages, signifying that individual

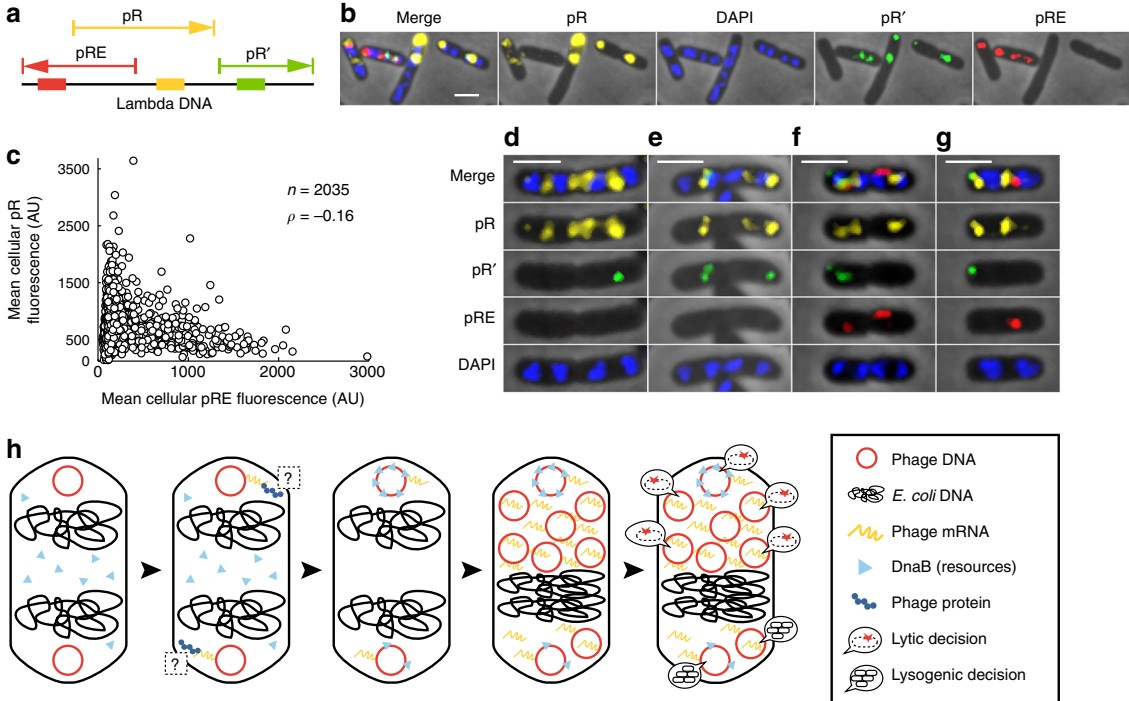

**Fig. 5 Individual phage decision-making occurs in separated subcellular locations. a** Colored arrows represent the transcripts targeted by FISH. The arrow direction corresponds to the transcription direction. The colored line segments on the black line mark the approximate locations of the FISH probes relative to the transcripts. **b** Decision-making transcripts localize to subcellular locations. Representative infected cells at 15 min post-infection with cells DAPI-stained. Representative cells chosen from four independent infection experiments. All scale bars in this figure are 2 μm. **c** Average pRE FISH signal is plotted against pR signal (Pearson's $\rho$ in plot, $p$ value <0.001). **d–g** Phages make different decisions in separate subcellular locations. pR transcripts, separated by DAPI clusters, occupy separate areas. Of 2035 cells, 645 have pRE foci. Of 2035 cells, 439 have pR′ foci. In all, 504/2035 cells have pRE without pR′ foci. Of 2035, 298 cells have pR′ without pRE foci. **d–e** Representative cells showing lytic decisions in a subset of different locations (75/2035 cells). **f–g** Representative cells showing conflicting lytic/lysogenic decisions in different locations (48/2035 cells). Representative cells chosen from four independent infection experiments. In total, 141/2035 cells have both pRE and pR′ in the same cell at any locations. **h** Model of lambda decision-making in subcellular space. Phage DNAs occupy different subcellular areas, separated by bacterial DNA. Phage DNAs undergo gene expression in their locations, consisting of transcription and translation. It is unknown where key proteins localize after detaching from mRNA. Phage DNAs sequester essential DNA replication resources to their own locations. Phage DNA replication transpires where individual DNAs are, and continued transcription remains localized with phage DNA. Individual units remain separated by bacterial DNA and can differ in composition. Expanding phage DNAs physically push bacterial DNA inside the cell. Individual decisions may be enacted by segregated phage DNAs. Source data are provided as a Source Data file.

lambda DNAs are capable of heterogeneous development in single cells[12,13]. Given our conclusions up until now, we would expect that these voting behaviors occur in separate subcellular areas. Accordingly, we found that in our FISH experiments, pR′ transcripts can exist in a subset of intracellular pR clusters (Fig. 5d, e). We also found that pRE and pR′ transcripts can coexist in different locations in single cells, suggesting that one intracellular phactory might vote differently from a neighboring phactory (Fig. 5f, g). The organization underpinning these behaviors persists even after new transcription initiation is blocked (Supplementary Fig. 15f–k, m). These data indicate that multiple intracellular lambda DNAs can execute divergent developmental pathways, where the spatial organization of phage/bacterial biomolecules and processes supports their individuality (Fig. 5h).

We described how multiple aspects of lambda development coalesce via spatial organization, allowing multiple viruses to develop separately in a cell. We characterized how the single phage DNA collects DnaB, essential for DNA replication, to its own location, begetting its clones and their corresponding development to its own coordinates. We showed that phage DNAs organize multiple transcripts locally at early time points, suggesting that different gene expression patterns are enacted by separate sets of viral DNA in separate cellular spaces, altogether

resulting in individual, subcellular compartments of phage development, or phactories. The individual phactory is formed both due to the phage's own actions, including DnaB recruitment and phage mRNA organization from active and finished transcripts, and also due to the physical separation of the viral biomolecules by the bacterial chromosome. It is unclear to what degree phage proteins are separated during the infection cycle, which is another important element contributing to divergent development in single cells (Fig. 5h).

It is intriguing that even simpler viruses, such as lambda, have evolved their own means of organization, which allows relatively complex behaviors to be achieved. When compared to particularly complex phages, such as *Pseudomonas* phage 201phi2-1, with a genome over 300 kbp encoding over 400 putative genes, lambda is simplistic[34]. 201phi2-1 houses its own cytoskeletal genes that position and build a protein-enclosed compartment encapsulating its DNA replication and transcription processes separately from its translation, evoking comparison to a nucleus[8]. Simpler phages, such as phage phi29, may have increased reliance or exploitation of host-specific proteins and processes. Efficient phi29 DNA replication relies on interactions with the host's MreB cytoskeleton to position phage DNA and replication machinery[35]. Phi29 DNA replication also depends on the organization of a replication resource, phage DNA polymerase, via the

phage-driven actions of its terminal protein[36]. Phage lambda has a similar strategy to organize replication, as it both utilizes its own proteins to actively reorganize its essential resource, DnaB, and also depends on its host's architecture, the location of the bacterial nucleoid. The relative simplicity of lambda can also be compared to some high-copy number plasmids lacking partitioning systems, which tend to localize to non-nucleoid locations, perhaps passively[37]. There are major differences between lambda and plasmid replication. Our data demonstrate that lambda retains its DNA into a single phactory, which pushes the nucleoid as it grows in space. This would obviously be detrimental for plasmids, because a single plasmid cluster in one cellular position would become easily lost during cell division, and if plasmids interfered with bacterial DNA localization, it would disrupt cell viability. Additionally, lambda actively commandeers DnaB, which clearly diverges from plasmid behavior, since this would kill the cell.

The application of high-resolution methods to phage lambda has the potential to reveal distinct and novel mechanisms buried within the long-standing lambda paradigm and could be equally applicable in other phage systems for comparison[38]. Further investigations into the biophysical interplay of viral and bacterial biomolecules, especially those which leverage continually advancing high-resolution techniques capable of fine spatial discrimination, promise new insights into the varied mechanisms behind viral development.

## Methods

**Strains, plasmids, and primers**. Bacterial strains, phages, and plasmids used in this study are listed in Supplementary Table 1.

Primers and homology regions used in this study are listed in Supplementary Table 3.

**Single-cell infection imaging assay**. Host cells, LZ1557, were grown from a −80 °C frozen permanent stock in 1 ml of M9 + 0.4% maltose (M9M), supplemented with antibiotics, 100 μg/ml ampicillin (Amp100) + 50 μg/ml kanamycin (Kan50) + 10 μg/ml chloramphenicol (Cm10) in a shaker at 37 °C, 265 r.p.m., overnight for ~24 h. This overnight culture was then diluted 1:1000 into 5 ml of M9M + antibiotics as above and grown under the same conditions for ~16–18 h, to $OD_{600}$ ~0.3–0.4. Then, 1 ml of this culture was pelleted in a tabletop centrifuge, $3000 \times g$ for 4 min at room temperature. During this time, 20 μl of purified reporter phage (λLZ1576) at ~3–4 × 10^10 pfu/ml is pipetted into a microcentrifuge at room temperature. After the cells were centrifuged, the supernatant was pipetted away, and the pellet was resuspended in 200 μl of room temperature M9M. Twenty microliters of this suspension was then mixed with the 20 μl of phage solution via gentle pipetting, resulting in an average phage input (API, the phage:bacterium ratio) of ~4, and then 80 μl of room temperature M9M was added to the mixture and mixed via gentle pipetting. This new mixture was then moved to a pre-warmed 35 °C water bath for 4 min to allow for phage adsorption and DNA ejection. During this time, a small (1–1.5 cm²) section from a room temperature M9M agarose pad (all pads in all experiments were 1.5% agarose), freshly made, was set onto a small No. 1 coverslip (18 × 18 mm). Following the incubation of the mixture, 1 μl of the mixture was deposited onto the M9M pad. After the mixture visibly dried, ~1 min, a larger No. 1 coverslip (24 × 50 mm) was overlaid onto the M9M pad, sandwiching it, and the sample was moved to the microscope for time-lapse imaging at 30 °C.

For the experiment using bacterial strain LZ1643 and phage λLZ1629, the M9M is supplemented with Amp100 + Kan50 for the bacterial growth. A colony from a plate was grown overnight for ~16–18 h. This overnight culture was then diluted 1:100 into fresh media and grown until $OD_{600}$ ~0.3–0.4, about 3–4 h. The cells were then pelleted and resuspended as described above. In these experiments, after the phage and cell suspensions were mixed, the mixture was set on ice for 30 min to pre-adsorb the phages before moving the mixture to the 35 °C water bath for 5 min. The following steps up to imaging are identical to the above description.

**Induction imaging assay**. A lysogen colony, LZ1596, from a plate was grown overnight in 1 ml of LB + 10 mM MgSO₄ (LBM) supplemented with Amp100 + Cm10, ~16–18 h in a 30 °C shaker, 225 r.p.m. The overnight culture was then diluted 1:100 into 5 ml of fresh LBM with the above antibiotics and grown under the same conditions until $OD_{600}$ ~0.3–0.4. For the 0 min time point, 1 μl of the culture was taken from the flask and deposited onto a phosphate-buffered saline (PBS) agarose pad, similarly as described above with the M9M pad, and then imaged. The lysogen culture was then induced by moving the flask to a 42 °C (our phages bear the $cI_{857}$ temperature-sensitive allele[39]), 225 r.p.m. shaking water bath

for 15 min. The 5-min time point occurs after 5 min at 42 °C, and the sample was processed and imaged at that time, as the 0 min sample was. The 15-min time point occurs at the end of the 42 °C incubation. While the 15-min time point sample was processed, the induction culture was moved to a 37 °C, 225 r.p.m. shaking water bath, and imaging of the 15-min time point occurred during this 37 °C incubation. The remaining time points were taken as the culture was shaking at 37 °C.

**Bacterial growth and lysogen induction assays**. To generate bacterial growth curves, bacterial strains were plated on a standard LB agar plate supplemented with appropriate antibiotics. A single colony was used to inoculate a 1 ml overnight culture in LB or M9 + 0.4% maltose (M9M) in a 37 °C, 265 r.p.m. shaker. The overnight culture was then diluted 1:100 into 30 ml of LB or M9 + 0.4% maltose (M9M) supplemented with appropriate antibiotics in a flask and grown in a 37 °C, 265 r.p.m. shaking water bath. $OD_{600}$ was measured using a spectrophotometer.

For lysogen induction assays, a lysogen colony from an LB plate was grown overnight in 1 ml of LB + 10 mM MgSO₄ (LBM) supplemented with appropriate antibiotics in a 30 °C, 180 r.p.m. shaking water bath. An overnight culture was then diluted 1:100 into 25 ml of fresh LBM with appropriate antibiotics in a flask and grown under the same conditions until an $OD_{600}$ of ~0.3, which serves as the 0-min time point. The lysogen culture was then transferred to a 42 °C, 180 r.p.m. shaking water bath for 15 min for thermal induction. The $OD_{600}$ of the culture was measured at the end of the 42 °C incubation, serving as the 15-min time point. The induction culture was moved to a 37 °C, 180 r.p.m. shaking water bath. The $OD_{600}$ of the culture was measured every 5 min until lysis with an $OD_{600}$ of ~0.05 using a spectrophotometer. Subsequently, 5 ml of the induction culture was taken from the flask and mixed with chloroform to a final concentration of 2% (vol/vol). The culture was agitated using a nutator for 15 min at room temperature and then centrifuged at $3000 \times g$ for 10 min to obtain the phage lysate. The concentration of the phage lysate was determined via a standard phage titration assay.

**DNA FISH**. For DNA FISH, probes for lambda DNA were produced by PCR amplifying ~3 kbp of the lambda genome (f-lambda-dnafish and r-lambda-dnafish primer pair), using a phage lysate as the template, and treating the purified PCR product with a PromoFluor500-dUTP nick translation kit (PromoCell) to generate DNA-PromoFluor500 fragments ranging from 100 to 500 bp. To generate probes for the E. coli attB region, an ~3 kbp region of the E. coli genome, including the attB region, was amplified with PCR (f-attb-dnafish and r-attb-dnafish primer pair) and treated with a PromoFluor640-dUTP nick translation kit (PromoCell). Equal amounts of these probes were mixed together to form a probe mixture.

To perform DNA FISH on infection samples, cells (MG1655) were first grown from a colony overnight, ~16–18 h, in LBMM (LB + 0.2% maltose + 10 mM MgSO₄). The overnight was then diluted 1:1000 into 50 ml of fresh LBMM and grown at 37 °C, 265 r.p.m., until $OD_{600}$ ~0.3–0.4, about 3–3.5 h. The culture was then pelleted via tabletop centrifuge ($2000 \times g$, 4 °C, 15 min), the supernatant was discarded, and the pellet was resuspended in LBM at 1/10th the original volume to concentrate the cells. Four milliliters of the cells were placed on ice, and ~40 μl of λLZ613 phage, at ~1 × 10^11 pfu/ml, was added to the cells and gently mixed. 2 × 500 μl aliquots of cells were also separated as a control without phages. After leaving the infection mixture for 30 min on ice, the tube was moved to a 35 °C water bath for 5 min for phage DNA ejection. At this point, 500 μl of the infection mixture was aliquoted into a culture tube with 4.5 ml of LB + 0.2% glucose + 10 mM MgSO₄ (LBGM) for each time point, all tubes were then moved to a 30 °C shaker at 265 r.p.m. At any given time, a tube was taken and fixed by pouring the mixture into a 15 ml centrifuge tube with 550 μl of 37% formaldehyde. This tube was left to shake on a nutator for 30 min, and then centrifuged at $4000 \times g$ for 3 min to pellet the cells. The control sample was fixed after the 35 °C incubation.

Details of fixation, permeabilization, and hybridization are detailed in other studies[25]. Briefly, the fixed cells were washed with 1 ml of ice-cold 1× PBS three times and resuspended in 1 ml of GTE solution (50 mM glucose, 20 mM Tris-HCl [pH 7.5], 10 mM EDTA). For the control sample, three separate 500 μl aliquots of the cell suspension were then mixed with 10 μl of 0.01 μg/μl lysozyme solution and incubated at room temperature for 2, 4, and 6 min followed by three washes with GTE, pelleting the cells via centrifugation at $10,000 \times g$ for 30 s. The cells were then resuspended in ~150 μl of GTE. For each control sample, 1 μl of the cells was deposited onto a PBS agarose pad and imaged. The lysozyme treatment time yielding ~90–95% intact cells (~1–5% lysed cells) represents the optimal treatment time for the samples. The actual time point samples, from the initial GTE wash, were then processed as the control was, using the optimal lysozyme time. For each time point, 10 μl of cells were deposited onto poly-L-lysine-coated large coverslips (24 × 50 mm), then covered with a smaller, normal, coverslip (22 × 22 mm). The coverslips were then immersed in 1× PBS and the smaller coverslip was removed, leaving only the sample coverslip. The cells were then dehydrated by immersing the coverslip in increasing concentrations of ethanol (70, 90, then 100%). Samples were then ready for hybridization.

For each sample, approximately 160 μg of the probe mixture was combined with 10 μl of hybridization solution (50% formamide, 10% dextran sulfate, 50 mM NaPO₄/pH 7, 2× SSC). The dsDNA probes were denatured at 75 °C in a thermocycler, then placed on ice. Ten microliters of the denatured probe mixture were then deposited onto the center of the sample on the coverslip and overlaid with a small coverslip (22 × 22 mm). The small coverslip was then sealed with nail

polish, forming a sample chamber. The chambers were incubated at 80 °C for 5 min to denature the cellular DNA, and then placed on Kimwipes over ice for 5 min. The chambers were then incubated in a 37 °C incubator overnight to complete hybridization.

The next day, the chambers were immersed in 2× SSC until the smaller coverslip dislodged. The remaining coverslips were soaked in wash solution (2× SSC, 50% formamide) for 20 min at 37 °C twice. The coverslips were then washed with a series of increasing SSC concentration washes (1, 2, then 4×), each for 5 min at room temperature. A DAPI solution was then made by mixing 1 μl of 10 mg/ml DAPI to 1 ml of 4× SSC. For each sample, 500 μl of the DAPI solution was added over the sample, covering it, and incubated for 5 min at room temperature. After drying the coverslip, 10 μl of 2× SSC was added over the sample and overlaid with a small coverslip (22 × 22 mm). The samples were then imaged.

**RNA FISH**. Different probes were synthesized to target different phage transcripts (Biosearch Technologies). Probes targeting pR and pRE were designed following previous studies[23,24], labeled with Cy5 and TAMRA, respectively. Probes targeting pR′ followed the same design principles as pR and pRE, and were labeled with AlexaFluor488 (pR′ probes listed in Supplementary Table 2).

To perform RNA FISH, we follow the same infection protocols as described above for DNA FISH. At given time points, the cells were fixed in formaldehyde and pelleted. In one set of experiments, samples were taken between 6 and 40 min, after infection by phages at ~2 × 10^10 pfu/ml. In another set of experiments, samples were taken at 15 min, after infection by phages at ~1, 2, 3, and 4 × 10^11 pfu/ml. The processing of the samples is detailed in our previous study[24]. Briefly, after fixation, the cells were washed three times with 1× PBS. Subsequently, the cells were permeabilized by resuspension in 70% ethanol for 1 h at room temperature and centrifuged to collect the cells. The pellet was then resuspended in wash solution (40% formamide, 2× SSC) and incubated for 5 min at room temperature, and pelleted again, ready for hybridization.

The cells were then resuspended in 25 μl hybridization solution (40% formamide, 2× SSC, 1 mg/ml E. coli tRNA, 2 mM ribonucleoside-vanadyl complex, and 0.2 mg/ml BSA) with each set of probes reaching a final concentration of 1 μM. The samples were then incubated in a 30 °C water bath overnight. The next day, the cells were washed three times using wash solution by incubating the cell pellet for 30 min in a 30 °C water bath. After the final wash, the cells were resuspended in wash solution + 10 μg/ml DAPI and incubated for 10 min at room temperature. This suspension was then pelleted and resuspended in 2× SSC. The sample was then ready for imaging.

For the infections with rifampicin, rifampicin was added at a final concentration of 50 μg/ml to a 50 ml infection mixture in a flask at 15 min after the 35 °C step. Instead of pre-aliquoting separate infection tubes, 5 ml of the infection mixture was withdrawn at each given time point after addition of rifampicin for fixation and further processing as described.

**E. coli nucleoid imaging after rifampicin treatment**. Bacterial cells (MG1655) were inoculated from a colony into 1 ml of LB and grown at 37 °C, 265 r.p.m. for overnight. The overnight culture was then diluted 1:1000 into 5 ml of LB and grown under the same conditions until an OD_600 of ~0.3. To study the effect of rifampicin on E. coli nucleoid morphology, rifampicin was added to the culture aliquots at final concentrations of 50, 100, and 300 μg/ml, and treated the cells for 15 min and 30 min at 37 °C, 265 r.p.m. One milliliter of the cell cultures was then fixed in 3.7% formaldehyde for 30 min at room temperature, followed by a washing step with 1 ml of PBS. The cell pellet was resuspended in 100 μl of PBS to reach an optimal cell density for microscopy imaging. For nucleoid imaging, 10 μl of cells were mixed with 10 μl of 20 μg/ml DAPI (to reach the final concentration of 10 μg/ml DAPI) for 10 min at room temperature. One microliter of the DAPI-stained cells were spotted onto a PBS agarose pad and imaged.

**SeqA cell lysogenization culture**. An overnight culture of cells bearing the SeqA reporter (LZ1557) were diluted 1:1000 and grown in 10 ml of LBMM + Kan50 + Amp100 + Cm10 to OD_600 ~0.4. The culture was centrifuged, and the pellet was resuspended in 1 ml of LBM to concentrate the cells by 10-fold. Two hundred and fifty microliters of cells were mixed with phage (λLZ1576) to reach an API of ~4. The infection mixture was placed on ice for 30 min, and then moved to a 35 °C water bath for 5 min. The mixture was then diluted into 5 ml of fresh LBM and incubated in a 30 °C shaking water bath at 265 r.p.m. Samples were withdrawn at given time points for imaging.

**Microscopy imaging**. Each set of experiments (live-cell with λLZ1576/LZ1557, live-cell with λLZ1629/LZ1643, lysogen induction, DNA FISH, and RNA FISH) had its own set of imaging parameters according to the specific strains and fluorophores employed. All imaging was performed on a Nikon Eclipse Ti inverted epifluorescence microscope using a 100× objective (Plan Fluo, NA 1.40, oil immersion) with a 2.5× TV relay lens, using a mercury lamp as the light source (X-Cite 200DC, Excelitas Technologies), within a cage incubator (InVivo Scientific) at 30 °C, and acquired using a cooled EMCCD (electron multiplying charge-coupled device) camera (iXon3 897; Andor, Belfast, United Kingdom). The software images

each stage through each filter sequentially for each time point before moving to the next stage. For the induction and fixed-cell experiments, stages with abundant cells were chosen for imaging. The stages were imaged under phase-contrast and specific filter cubes. The fluorescent filters used in the study were as follows (X, Y [excitation bandwidth] excitation filter/dichroic beamsplitter wavelength/X, Y [emission bandwidth] emission filter/company, product #): DAPI (350 nm, 50ex/400 nm/460 nm, 50em/Nikon, 96310), blue (436 nm, 20ex/455 nm/480 nm, 40em/Nikon, 96361), custom green (490 nm, 20ex/505 nm/525 nm, 30em/Chroma, custom 49308), yellow (500 nm, 20ex/515 nm/535 nm, 30em/Nikon, 96363), orange (539 nm, 21ex/556 nm/576 nm, 31em/Chroma, 49309), Cy3 (545 nm, 30ex/570 nm/610 nm, 75em/Nikon, 96323), red (560 nm, 40 nm/585 nm/630 nm, 75 nm/Nikon, 96365), far red (592 nm, 21ex/610 nm/630 nm, 30em/Chroma, 49310), and Cy5 (615 nm, 70ex/660 nm/700 nm, 75em/Nikon, 96366).

For imaging each experiment, samples were exposed to the named filter cube in this order (with this exposure time, against this target).

Infection movies using λLZ1576/LZ1557: phase-contrast (100 ms), blue (1 s, DnaB), orange (100 ms, single phage DNA), far red (200 ms, replicated phage DNA), and green (40 ms, capsid). This cycle of imaging occurred automatically once every 10 min for at least 3 h in each movie.

Infection movies using λLZ1629/LZ1643: phase-contrast (100 ms), blue (100 ms, capsid), yellow (100 ms, attB), and red (200 ms, replicated phage DNA). This cycle of imaging occurred automatically once every 5 min for at least 2 h in each movie.

Lysogen induction: phase-contrast (100 ms), blue (1 s, DnaB), far red (200 ms, replicated phage DNA), and green (40 ms, capsid). This was not a movie, so the imaging cycle was repeated for each stage/time point in the dataset.

DNA FISH: phase-contrast (100 ms), yellow (200 ms, phage DNA), Cy5 (200 ms, attB), and DAPI (30 ms, DAPI). This imaging cycle was repeated for each stage in the dataset.

RNA FISH: phase-contrast (100 ms), Cy5 (200 ms, pR), Cy3 (200 ms, pRE), yellow (200 ms, pR′), and DAPI (30 ms, DAPI). This imaging cycle was repeated for each stage in the dataset.

E. coli nucleoid imaging after rifampicin treatment: phase-contrast (100 ms) and DAPI (100 ms).

**Data analysis**. Microscopy images were analyzed using the cell recognition program Schnitzcells (gift of Michael Elowitz, California Institute of Technology), the spot recognition program MicrobeTracker (gift of Christine Jacob-Wagner, Yale University), and homemade scripts in Matlab (Supplementary Discussion).

**Strain construction**. To construct the phages with tetO arrays, phages bearing fluorescent reporters (λLZ1269, λLZ1369, λLZ1527, details for constructing these types phages were reported in previous studies) served as the parents[13]. Then a tetO-recombination plasmid was constructed to replace the bor::Kan^R region of the fluorescent phages with a bor::Cm^R 24×tetO array construct, using homology regions (upstreambor and downstreambor) The tetO array was derived from another study[40], and inserted adjacent to a Cm^R cassette. Phages were titered onto host cells bearing the tetO-recombination plasmid (pBR322 24×tetO bor::Cm^R) and a pLate*D plasmid, then lysogenized the resulting plate lysate into MG1655, selecting for Cm resistance and Kan sensitivity in single-integration lysogens. The genomic construct was then verified by PCR.

To construct the triple reporter strain, bearing SeqA (single phage DNA), TetR (replicated phage DNA), and DnaB reporters, strain LZ1383 (MG1655 seqA-mKO2 Cm^R-FRT) served as the parent. Plasmid PCP20 was transformed into this strain to recombine out the Cm^R cassette flanked by FRT sites[41], imparting Cm-sensitivity to the cell (LZ1535). Then, MG1655 dnaB-mTurquoise2-Cm^R-FRT (LZ1510) was generated by first constructing the plasmid, pdnaB-mTurquoise2-Cm^R-FRT (H is the downstream homologous region of dnaB). Next, PCR was performed to create a linear dsDNA of the dnaB-mTurquoise2-Cm^R-H region, which was used for red-recombination[42], to generate LZ1510. Then P1 transduction[43] was performed to move the dnaB-mTurquoise2 reporter from LZ1510 (donor) to LZ1535 (recipient), making a strain with both the SeqA and DnaB reporters (LZ1552). Next, the Δdam-Kan^R marker was transduced from LZ1386 to LZ1552 to complete the SeqA reporter (LZ1555). This new strain was transformed with pACYC177 pFtsKi tetR-mCherry to complete the triple reporter strain, LZ1557.

To construct the attB reporter strain, MG1655 served as the parent, and a 96× lacO array was inserted via red-recombination. This was done by first inserting upstream (E. coli genome region amplified with the f-up-attb and f-up-attb primer pair) and downstream (E. coli genome region amplified with the f-down-attb and f-down-attb primer pair) homology regions flanking the 96× lacO Kan^R region of a plasmid[40], and then digesting this plasmid (pattB 96× lacO Kan^R) within the homology regions to produce linear dsDNA for red-recombination. This strain was then transformed with pACYC177 pFtsKi tetR-mCherry lacI-eyfp.

To construct the lysogen with DnaB, TetR, and gpD reporters, LZ1510 (dnaB-mTurquoise2 Cm^R-FRT) served as the parent. The Cm^R cassette was removed using PCP20 to generate LZ1511 (dnaB-mTurquoise2) and then transformed pACYC177 pFtsKi tetR-mCherry into the strain. This host was then lysogenized with λLZ1575 (cI_857 D-mNeongreen bor::Cm^R 24×tetO), producing LZ1596.

**Phage purification**. The purification protocol was adapted from other sources[12,44,45]. Briefly, a single colony of desired lysogens was grown with appropriate antibiotics at 30 °C overnight. The overnight culture was then diluted into 500 ml and induced. The phages were then precipitated using 10% PEG8000 + 1 M NaCl. The resulting phage pellet was soaked in a total of 8 ml of cold SM buffer and incubated at 4 °C overnight, ~16 h. An organic extraction was performed by mixing the SM suspension gently with an equal volume of chloroform and centrifuging at 3000 × g for 15 min at 4 °C. The supernatant was removed to exclude the PEG pellet, and the extraction step was done two more times, to finally yield a clear supernatant containing the phage. A step gradient was made for each desired phage using 1.5 ml each of 1.3, 1.5, and 1.7 g/ml CsCl + SM buffer solutions, and the phage (~8 ml) was layered on top in a 13.2 ml ultraclear tube (Beckman Coulter), then ultracentrifuged in a Beckman SW41Ti rotor at 24,000 r.p.m. for 6–8 h at 4 °C. The phages migrated to a band and were then extracted from the side wall of the tube. This phage extraction was then loaded into a 5 ml ultraclear tube (Beckman Coulter) and then filled with a 1.5 g/ml CsCl + SM buffer and ultracentrifuged in a Beckman SW50 rotor at 35,000 r.p.m. for 24 h at 4 °C, and then was extracted in the same manner. This new phage extraction was then dialyzed 1:1000 against SM buffer over ~24 h. This phage solution was then extracted and stored away from light and at 4 °C to be used in the experiments.

**Reporting summary**. Further information on research design is available in the Nature Research Reporting Summary linked to this Article.

## Data availability
All strains and raw experimental data are available from the corresponding author upon request. Source data are provided with this paper.

## Code availability
Data analysis scripts are available from public repositories at https://doi.org/10.5281/zenodo.3906930.

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

## Acknowledgements
We are grateful to Craig Kaplan and Ryland Young for commenting on the earlier versions of the manuscript. We thank David J. Sherratt for gifting strains. This research was supported by National Institutes of Health (grant R01GM107597), National Science Foundation (grant 2013762), and Texas A&M University X-Grant (grant 290386) to the Zeng Laboratory.

## Author contributions

J.T.T. and L.Z. conceived the project. J.T.T., Q.S., J.G., and L.Z. designed the experiments. J.T.T. and J.G. constructed strains. J.T.T. performed the live-cell imaging experiments. Q.S. and J.G. performed the RNA FISH experiments. J.G. performed the DNA FISH experiments. Q.S. and J.T.T. built the data analysis framework. J.T.T., Q.S., and J.G. analyzed the data. J.T.T., J.G., and L.Z. wrote the paper. L.Z. supervised the project.

## Competing interests

The authors declare no competing interests.
