## [Peer Review File · Nature Communications]

Reviewers' comments:

Reviewer #1 (Remarks to the Author):

Trinh et al use a series of fluorescent labels in order to keep track of virus DNA and its replication, transcription of key lambda genes that track the decision of lysis/lysogeny, and assembly of capsids and hence number of virions. They follow the process of infection in real time in single cells and in some cases with single molecule resolution. Certain experiments are performed in situ and hence not in real time. Based on a large set of measurements they find that different infecting DNAs occupy different subcellular compartments and interpret these results as a form of individuality.

I have several issues with the presentation and the results:

1. The manuscript is hard to read as it is not subdivided in sections with subtitles that would make the storyline and arguments more convincing.
2. The supposed individuality they see could be simply a result of looking at cells that are not synchronized in terms of cell cycle.
3. It is not clear whether the fluorescent labels they use do not cause aggregation and thus artefacts. There is a famous paper from Johan Paulsson lab at Harvard which attest to such issues of aggregates. (Landgraf et al Nat Methods 2012)
4. The interpretation of individuality has no consequence on the nature of individual phages as they are all the same after all, so I do not see any biological relevance. Many things are stochastic in cells yet not of biological relevance.

Furthermore, in order to see how much the nucleoid itself modulates what the authors see, I would suggest looking at minimal media grown cells versus rich media grown cells, where nucleoid dynamics and compaction vary dramatically and to decorate the nucleoid fully (eg Fis-GFP John Marko lab has published work that can be of help for such experiments, Yazdi et al Mol Microbiol 2012) such that we can clearly see what spaces are free.

I also have a hard time with the many anthropomorphic terms. A factory relies on input of energy, planning, and much more than self-assembly. To talk about “communities” of self-assembling molecules I also find misleading. Just by introducing new terminology to effects that have already an old nomenclature does not make for new science. I know that there is a trend of finding new catch words, but this has really been abused.

Throughout the manuscript it is never clear when we look at single infections or multiple infections, thus some basic aspects are not clear. Also throughout it is completely not clear what is cause and what effect, what we see is due to bacterial biology or to some intrinsic phage mechanism? Is it happenstance or is there a mechanism that has some consequence/advantage that it confers to the virus on a global level or individual one. What are the implications of individuality if any?

Based on the above issues, I think that the work is too preliminary for Nature Communications, as the conclusions are not that solid, despite a lot of data and the relevance not clear.

Reviewer #2 (Remarks to the Author):

A combination of sophisticated techniques allows Trinh et al. to reach conclusions on the compartmentalization of phage lambda replication and transcription in E. coli cells. Demonstrations are mostly convincing and supported by large bodies of data and image analyses. I have no major criticism on this impressive work, for which I want to congratulate the authors.

Being an expert in phage physiology rather than single cell analyses, my minor comments will only aim at making the text more accessible to a large readership.

1. Lines 82 and 159: One would like to relate these single cell experiments to classical experiments. At which MOI are the experiments done? From my calculations it is roughly 100, and at such high MOI, one expects a large fraction of lysogenization events. Is it the case? Or does the CI857 allele + temperature choice (35) prevent lysogenization?
2. How long does the lytic cycle takes, under this minimal medium, 35°C growth conditions? At what time point(s) lysis is seen?
3. The dnaB fusion is a new construct. How viable are the strains with this dnaB-mTurquoise fusion compared to the MG WT, and how does lambda replicate in such a strain (burst size, latency)?
4. Same question with the lambda with a tetO array: does it grow fine, compared to its parent phage? Sometimes such arrays tend to block replication, or make it stumble.

5. Lines 92-94: why choosing this 'DnaB sequestration' interpretation, rather than an accumulation of replicating forks at the phage factory?
6. Line 99: the TetR clusters 'above the TetR background', that represent the bulk of replicated phage DNA, are not easy to visualize by eye in fig1e, g, movie3 (it may be also due to picture size). One understands (with suppl. text) that this is due to the fixing of a given brightness for the whole movie. Suppl. Fig6 (where the brightness is not fixed) helps a lot seeing these clusters, and I would recommend this figure to be shifted into the main text (see also below point 8).
7. Line 105: given the large surface of the TetR clusters at the time where GpD labeled capsid arise, it is not surprising that the two signal mostly overlap. Can you, under such conditions, really suggest 'that the locations of phage DNA determine where progeny phages are assembled'? I'd rather skip this sentence.
8. Lines 109-112: 'a single phage DNA organizes its own subcellular factory', this sentence sums up a main point of the paper, it needs to be more precise : this organization takes place only at early time points, right? This is an important facet of the observations. From the perspective of homologous recombination (which is quite efficient between phage genomes), one would not understand how and when it takes place if all 'phactories' remain separated until the end.
9. Line 115. The visual demonstration of various separated factories is much more convincing in suppl. Fig6 than in Fig1g. Consider adding Suppl. Fig6 in the main text. It is a different set-up, with lysogenic induction, so it strengthens the point.
10. What is a 'cache'?
11. Line 119: 'where the specific make up of the communities may change and be exchanged while still maintaining spatial organization'. This sentence is vague, what is the 'specific makeup'? do you mean the phage proteins are exchanged? How do you know that?
12. Line 140: 'attB near the quarter-cell'. According to the scheme below fig2h, quarter cell should correspond to a value of 0.5, whereas the peak of attB location is at 0.25. So more near mid-cell?
13. Line 150: a word seems to be missing before 'provide'
14. Line153: Please mention on what result is drawn the conclusion 'the physical barrier allows subcellular viruses to maintain different entities'? Is it on the 'squeeze' cases?
15. Line 159: one has to dig into the methods section to learn with which lambda phage the FISH experiments were done, please insert 'we labeled LZ613 phage DNA...'
16. Line 169: the sentence 'we observed that the spatial distribution of the E.coli leaves..', is unclear, do you mean 'of the E. coli nucleoid'?
17. Paragraph lines 106-118: the offset of pRE and pR signals is not super evident from the suppl fig. 14e and f.

18. Line 225: what is the proportion of cells with pRE foci? Among them, how many also have pR' foci? Does it fit with lysogenization frequencies? The underlying question is 'does it need a pRE-only cell to form a lysogen'?

19. Line 233-234: 'we showed that phage DNAs organize their transcripts locally' please complete the sentence with 'at early time points', or 'at 15 min'.

20. Suppl. text, page 2, 6 lines from bottom, 'for example in Fig1e, 70 min', there is no 70 min time point in this figure. Is it 80min?

21. Suppl. Fig11, title of the legend: 'DNA and which', remove 'and'

22. Suppl. Fig11, legend b-d. The lytic cells' data are probably from Fig. 2h, rather than 3h.

Reviewer #3 (Remarks to the Author):

In this paper Trinh and co-workers have used a number of quantitative microscopy techniques to investigate the intracellular spatial organization of *E. coli* during phage infection. They have imaged multiple stages of the infection cycle using an impressive number of different fluorescent fusions/probes. The work is of good quality and overall offers an interesting new insight into stochastic processes in bacteria with a unique observation of heterogeneity within individual *E. coli* rather than between cells. I have listed my comments below; my only major concerns are firstly regarding the fusions; given the somewhat unexpected behavior of some of them (particularly DnaB) the authors need to do more to verify the tagged proteins remain fully functional. Secondly, the authors need to more to quantitatively verify that the phactory concept they describe is responsible for the divergent phage gene expression they see with FISH.

- Could the authors elaborate why their DnaB fusion does not form foci in normal growing cells as would be expected from ongoing chromosomal replication in these conditions? I would imagine replication foci should be visible given the high-quality microscope and EMCCD camera used in this study: for example, a recent paper on *E. coli* replication dynamics after UV damage a YPet-DnaB forms clear foci under normal growth conditions (Soubry N, Wang A, and Reyes-Lamothe R (2019) Replisome activity slowdown after exposure to ultraviolet light in *Escherichia coli*. PNAS 116(24):11747-11753). Was this fusion verified by sequencing, and does this fusion have the same growth rate and average cell size to the WT? If possible, I would also suggest flow cytometry to verify the number of chromosomes per cell is the same as WT. Can they rule out having selected for a gene duplication with an unlabeled DnaB copy during the red-recombination?

- page 5 line 95: 'time points' is not a very useful measure – please use minutes.

- The authors conclude that the *E. coli* chromosome provides a barrier which determines phactory location. However, in the later FISH experiments they use rifampicin treatment and find that phactory organization appears to remain. Rifampicin causes a rapid loss of nucleoid organization and chromosomal DNA fills the whole cell volume after treatment (See Cabrera, J. E., & Jin, D. J. (2003). The distribution of RNA polymerase in *Escherichia coli* is dynamic and sensitive to environmental cues. *Molecular Microbiology*, 50(5), 1493–1505). If the nucleoid was the primary barrier in phactory formation I would have hypothesized that rif treatment would lead to a loss of phactory organization. From the images in Supplementary Fig. 15a-e, it appears that the nucleoid structure is retained after rif treatment used, so perhaps the concentration is not high enough to see this effect. In the attB experiments I would suggest incubating with a higher concentration of rif which does causes nucleoid structure loss (this can be verified by DAPI), to see if bacterial and phage DNA separation is lost. This would help verify their hypothesis.

- The distinct gene expression profile between phactories in the same cell is an important aspect of this work, but the authors need to do more to quantitatively show that the compartmentalization they describe is responsible for this. In Fig. 4, panel f,g, representative cells are shown from 48/2035 cells where different compartments have exclusively pR' or pRE signal. What percentage cells have both signals in the same compartment? For cells which have both signals are they significantly more likely to be located in different compartments? How does the heterogeneity in expression between same cell compartments compare to between cells? Are more spatial distant compartments more likely to make different decisions? The authors could try using subinhibitory concentration of cephalixin prior to infection to generate filamentous cells containing many separate nucleoids (for examples see Nonejuie et al (2013). Bacterial cytological profiling rapidly identifies the cellular pathways targeted by antibacterial molecules. *PNAS*, 110(40), 16169–16174). These should be enriched for single cells making multiple decisions.

- In general, the retention of phage RNA near its site of origin even after rifampicin treatment is very interesting - do the authors speculate that this behavior is unique to phage mRNA, or would host mRNA be similarly confined within the same 'compartments'? If this is simply caused by entropic forces preventing 70S ribosomes diffusing through the nucleoid (as seen by Sanamrad et al (2014). Single-particle tracking reveals that free ribosomal subunits are not excluded from the *Escherichia coli* nucleoid. *PNAS*) then non-phage mRNA would also be expected to be similarly compartmentalized. This would be interesting to look at in future work.

Detailed Responses to Reviewers' Comments

Reviewer #1

We thank the reviewer for reading our work and commenting on it. We appreciate the recognition of our “large set of measurements” and varied methods. The reviewer had issues with some aspects of the work, which we addressed as appropriate. We believe our clarifications will help alleviate concerns and improve the understanding of our work.

1. The manuscript is hard to read as it is not subdivided in sections with subtitles that would make the storyline and arguments more convincing.

Response: We have divided our writing into clear subsections. Thanks for the suggestion.

2. The supposed individuality they see could be simply a result of looking at cells that are not synchronized in terms of cell cycle.

Response: We agree that different cells may be at different stages at a given time point in our images despite our efforts to synchronize them as described in the Methods. We do not believe that this undercuts any of our conclusions. Our model is phage-driven, and the phage will develop regardless of the status of the cell cycle. We showed that the phages develop in individual compartments across different cells, which encompass different states of the cell cycle. Therefore, our claims of individuality are independent of the cell cycle.

3. It is not clear whether the fluorescent labels they use do not cause aggregation and thus artefacts. There is a famous paper from Johan Paulsson lab at Harvard which attest to such issues of aggregates. (Landgraf et al Nat Methods 2012)

Response: This is a cogent point, as spurious aggregation of fluorescent proteins could confound our conclusions. We do not believe this occurs in our work. Our constructs have either been validated previously (e.g. TetR reporter: Lau, IF, et al. Mol Microbiol. 2003; Wang, X, et al. Genes Dev. 2005. SeqA reporter: Babic, A, et al. Science. 2008; Shao, Q, et al. Biophys J. 2015; Shao, Q, et al. MicrobiologyOpen. 2016; Trinh, JT, et al. Nat Commun. 2017. gpD reporter: Zeng, L, et al. Cell 2010; Trinh, JT, et al. Nat Commun. 2017) or we have validated in this work through extensive control experiments (e.g. DnaB construct, the new Supp. Fig. 2). We have also included FISH experiments which do not involve fluorescent proteins to support our data in the manuscript.

4. The interpretation of individuality has no consequence on the nature of individual phages as they are all the same after all, so I do not see any biological relevance. Many things are stochastic in cells yet not of biological relevance.

Response: We respectfully disagree that individuality has yet not of biological relevance. It made a huge difference whether phages act as a mass-action pool or phages as individual decision makers, which lead to gene dosage mechanism or individual phage voting mechanism respectively. These differences were published in Zeng, L, et al, Cell 2010. In our previous work (Trinh, JT, et al. Nat Commun. 2017), we found interesting phage-phage interactions inside the cell including competition and cooperation which is another evidence of the interesting behaviors by individual phages, and in this work we elaborate the inner workings of how individual phages behave at the subcellular level in depth.

5. Furthermore, in order to see how much the nucleoid itself modulates what the authors see, I would suggest looking at minimal media grown cells versus rich media grown cells, where nucleoid dynamics and compaction vary dramatically and to decorate the nucleoid fully (eg Fis-GFP John Marko lab has published work that can be of help for such experiments, Yazdi et al Mol Microbiol 2012) such that we can clearly see what spaces are free.

Response: Indeed, different growth conditions can affect the physiology of the cell and the phage consequently. Our live-cell time-lapse infection experiments (Fig. 1, 3) were done using minimal media, our live-cell non-time lapse induction experiments (Fig. 2) and our fixed-cell imaging experiments (DNA FISH, Fig. 4; RNA FISH, Fig. 5) were done using rich media. We have varied our growth conditions and these experiments converge to the same conclusions.

6. I also have a hard time with the many anthropomorphic terms. A factory relies on input of energy, planning, and much more than self-assembly. To talk about “communities” of self-assembling molecules I also find misleading. Just by introducing new terminology to effects that have already an old nomenclature does not make for new science. I know that there is a trend of finding new catch words, but this has really been abused.

Response: We respect the philosophical differences on how to describe biological processes.

Specifically, regarding the comment on a phage “factory,” this is not a new term or way of thinking about phage biological processes. Long ago, researchers have questioned in different phage, the topology of phage gene products and hypothesized that localization of these enzymes could resemble a subcellular factory (Kozinski, AW and Kozinski, PB, PNAS. 1967). In this work, our data reflect restrictive position, co-localization, and recruitment of host resource for phage development, etc, so we feel that a phage factory can be used here. We termed as “phactory” using “ph” uniquely for phage.

On the comment about “communities,” we agree that perhaps this invokes more complexity than intended in this manuscript. We decided to use “microenvironment” instead, which may be a better fit in this work.

7. Throughout the manuscript it is never clear when we look at single infections or multiple infections, thus some basic aspects are not clear.

Response: The average MOI is ~4 for the infection experiments (added to the Methods). The individual MOI of each cell varies according to the Poisson distribution of phage adsorption to the cell surface. In this manuscript, we did not specifically determine the MOI of each cell since the driver of individuality is the individual phage DNA, not the infecting viral particle on the cell. Due to the high rate of failed infection (20-30% published in Zeng, L et al, Cell. 2010; Trinh, JT, et al, Nat Commun. 2017), MOI is not an accurate count of successfully injected DNA. In addition, the individual phage DNA inside the cell can be the injected phage DNA and/or replicated copies of the injected DNA. In other words, the conclusions for this study do not depend on the number of infections, only on the number and location of DNAs inside the cell at the early time point upon infection. We also drew the same conclusions from cells that were not infected at all, which underscores this point, using lysogenic induction to show the localized phage development (new Fig. 2).

8. Also throughout it is completely not clear what is cause and what effect, what we see is due to bacterial biology or to some intrinsic phage mechanism? Is it happenstance or is there a mechanism that has some consequence/advantage that it confers to the virus on a global level or individual one. What are the implications of individuality if any?

Response: We think it is a combination of bacterial biology and phage mechanisms. For example, we stated that the phage alters the natural behavior of DnaB, the phage maintains its clones and resources, the expansion of phage DNA was a phage-active mechanism, and the phages execute divergent developmental pathways. Phages are viruses, so they neither fully de-couple from, nor fully rely on the host.

We believe this work represents a point from which multiple projects can branch from, rooted in these findings that even identical phages can act as individuals in a cell. As is, the results are

consequential for providing mechanisms for earlier questions and establishing a new perspective to think about this, and other systems moving forward.

Reviewer #2

We would like to express our appreciation for Reviewer #2's careful reading of the manuscript. We are very happy that this reviewer considered our results to be "convincing" and "impressive". The reviewer had a few questions/concerns which we will do our best to address here.

1. Lines 82 and 159: One would like to relate these single cell experiments to classical experiments. At which MOI are the experiments done? From my calculations it is roughly 100, and at such high MOI, one expects a large fraction of lysogenization events. Is it the case? Or does the CI857 allele + temperature choice (35) prevent lysogenization?

Response: The average MOI is ~4 (added to the Methods). In our infection experiments under the microscope, we were using *dam*⁻ cells, grown at 30°C, on a M9 pad. Under these conditions, lysogenization is less frequent than in aerated LB conditions with wild type cells.

2. How long does the lytic cycle takes, under this minimal medium, 35°C growth conditions? At what time point(s) lysis is seen?

Response: For the microscope experiments with *dnaB-FP* and *dam*⁻ alleles, the lysis time is 249 ± 33 min (mean ± std) for the 49 cells which lysed within the time frame of the imaging.

3. The *dnaB* fusion is a new construct. How viable are the strains with this *dnaB*-mTurquoise fusion compared to the MG WT, and how does lambda replicate in such a strain (burst size, latency)?

Response: The strain with *dnaB-mTurquoise2* is as viable as MG1655, its parent. It has a similar doubling time of 26 min in LB. To test lambda replication, we did lysogenic induction of a *D-mNeongreen* phage in isogenic WT *dnaB* and *dnaB-mTurquoise2* backgrounds and found that the resulting titers and lysis times are similar. Therefore, we think that the DnaB construct does not detrimentally affect lambda or host growth. These experiments and data were described in Supplementary Text (In 247-296) and Supp. Fig. 2g-h, and referred to in the text (In 69-70).

4. Same question with the lambda with a tetO array: does it grow fine, compared to its parent phage? Sometimes such arrays tend to block replication, or make it stumble.

Response: We did lysogenic induction of a phage with a *tetO* array (LZ1576) along with an isogenic phage without *tetO* (LZ1527). The resulting titers are similar for both phages and the *tetO* version has a ~5 min delay in lysis time. Thus, the 24x *tetO* array may impose a minor reduction in replication rate with minimal effect on phage productivity. These data have been added to Supp. Fig. 2i.

5. Lines 92-94: why choosing this 'DnaB sequestration' interpretation, rather than an accumulation of replicating forks at the phage factory?

Response: We agree that "DnaB sequestration" can be misleading. We meant that DnaB is more concentrated where lambda has high level of replication due to more positive cooperative feedback for more DnaB localized. We have changed the wording to avoid confusion.

6. Line 99: the TetR clusters 'above the TetR background', that represent the bulk of replicated phage DNA, are not easy to visualize by eye in fig1e, g, movie3 (it may be also due to picture size). One understands (with suppl. text) that this is due to the fixing of a given brightness for the whole movie. Suppl. Fig6 (where the brightness is not fixed) helps a lot seeing these clusters, and I would recommend this figure to be shifted into the main text (see also below point 8).

Response: We have changed the image contrasts of Fig. 1 as the way we did in Supp. Fig. 2 for easier reading. We then moved the figure with static contrasts to Supp. Fig. 2b-e. We have also moved Supp. Fig. 6 to the new Fig. 2 in the main text.

7. Line 105: given the large surface of the TetR clusters at the time where GpD labeled capsid arise, it is not surprising that the two signal mostly overlap. Can you, under such conditions, really suggest 'that the locations of phage DNA determine where progeny phages are assembled'? I'd rather skip this sentence.

Response: We think it is reasonable to state this. The conclusion is consistent with the established biological mechanism of lambda virion assembly. The assembly of gpD onto virions occurs only as the phage procapsid is being packaged with DNA (Casjens, SR and Hendrix RW. J Mol Biol. 1974). The actual packaging of DNA exposes the binding sites for gpD on the capsid, so it is sensible that the location of phage DNA is where gpD will localize.

8. Lines 109-112: 'a single phage DNA organizes its own subcellular factory', this sentence sums up a main point of the paper, it needs to be more precise: this organization takes place only at early time points, right? This is an important facet of the observations. From the perspective of homologous recombination (which is quite efficient between phage genomes), one would not understand how and when it takes place if all 'phactories' remain separated until the end.

Response: We think the initial phage DNA can plant the seed for a phactory by gathering DnaB for replication, and the replicated DNA and subsequent newly assembled virions maintain localization. In our model, the organization takes place not just at early time points, but throughout the infection

We agree with the reviewer about the importance of accounting for recombination. We think the labelling method allows us to visualize concentrations of phage DNA, but this does not rule out diffusion of a fraction of the DNA out of the "phactory" zone (or compartment). It is important to note, however, that, in the absence of a constraining "pseudo-nucleus" structure like the one recently shown for the jumbo phage phiKZ (Chaikerasitak et al., Science 2017), the notion of the phactory does not preclude the diffusion of some individual DNA into other regions of the infected cell, perhaps to another phactory. Indeed, some exchange of progeny DNA may occur in cells with two or more infecting phages.

9. Line 115. The visual demonstration of various separated factories is much more convincing in suppl. Fig6 than in Fig1g. Consider adding Suppl. Fig6 in the main text. It is a different set-up, with lysogenic induction, so it strengthens the point.

Response: Yes. We have moved Supp. Fig. 6a-e and the associated text into the main text as a new Fig. 2. We left Supp. Fig. 6f-g as Supplementary.

10. What is a 'cache'?

Response: Cache means that each DNA binds to DnaB. To avoid the confusion, we have changed the wording (In 119).

11. Line 119: 'where the specific make up of the communities may change and be exchanged while still maintaining spatial organization'. This sentence is vague, what is the 'specific makeup'? do you mean the phage proteins are exchanged? How do you know that?

Response: It is true that this wording is lacking in detail for what we intended. To note, we decided to change "community" to "microenvironment" in this revised manuscript. In our model, a phactory is a subcellular compartment made up of phages (DNA) and their related gene products (RNA/proteins). In our experiments, we only used one genotype of phage, so different phactories are clonal, but they might vary in number of DNA, and number or identity of RNA and

protein. We made a change to specify this (ln 122-123). In our experiments, we can directly characterize that the levels of DNA vary, and that different RNAs exist in different factories. We do not know exactly where all of the proteins locate. At this resolution, using fluorescently labelled DNA and gpD protein, only capsid maturation was visualized, as gpD is assembled onto filled capsids (Casjens, SR and Hendrix RW. J Mol Biol. 1974).

12. Line 140: 'attB near the quarter-cell'. According to the scheme below fig2h, quarter cell should correspond to a value of 0.5, whereas the peak of attB location is at 0.25. So more near mid-cell?

Response: In our numbering scale for the figure (now Fig. 3h), the cell positions range from 0 (mid-cell) to 1 (cell pole), so 0 is the mid-cell and 0.5 is the quarter-cell. We changed our wording to say the peak is between mid-cell and quarter-cell (ln 145).

13. Line 150: a word seems to be missing before 'provide'

Response: Upon review, it appears correct to us. The sentence describes three actions regarding "data" in the list: the data agree, provide, and corroborate.

14. Line153: Please mention on what result is drawn the conclusion 'the physical barrier allows subcellular viruses to maintain different entities'? Is it on the 'squeeze' cases?

Response: Yes, to clarify, it is the "squeeze" scenario. It requires two or more factories in one cell to conclude that there is a barrier separating them. We added clarifying text (ln 158-159).

15. Line 159: one has to dig into the methods section to learn with which lambda phage the FISH experiments were done, please insert 'we labeled LZ613 phage DNA...'

Response: We have changed this (ln 166).

16. Line 169: the sentence 'we observed that the spatial distribution of the E.coli leaves..', is unclear, do you mean 'of the E. coli nucleoid'?

Response: We mean the DAPI signal labeling the nucleoid. We have clarified this (ln 176).

17. Paragraph lines 106-118: the offset of pRE and pR signals is not super evident from the suppl fig. 14e and f.

Response: This is true, that the way we presented this, there are no stark differences, but this correctly represents the subtle differences. This presentation of data is basically the full dataset of signals. RNA FISH foci/clusters are actually very bright, and the signal of the foci are almost all contained in the rightmost 3-4 boxplots. If comparing just those brightest data, it is more apparent that the pR signals in the pR' foci are brighter than the pR signals in the pRE foci. This is a non-biased method to measure the pR signal in the foci of pR' and pRE in each cell. This full-dataset figure complements Supp. Fig. 14h-i, where we defined foci in each cell. We can see here that the pR signals in pRE foci are subtly lower than in pR foci, per their distributions.

18. Line 225: what is the proportion of cells with pRE foci? Among them, how many also have pR' foci? Does it fit with lysogenization frequencies? The underlying question is 'does it need a pRE-only cell to form a lysogen'?

Response: We have reported lysogenization frequencies before (Zeng, L. et al. Cell 2010, Trinh, JT et al. Nat Commun. 2017, Shao, Q et al. Microbiologopen 2016, Shao, Q et al., iScience 2018) but would not draw strong conclusions about how FISH data match our lysogenization data. FISH represents a single snapshot in time, which differs greatly from other assays. In FISH, we also do not know about the important decision-making proteins, as we did not perform immunofluorescence with FISH. In other words, cells with pRE transcripts may or may not have

an abundance of CI protein and cells with pR' transcripts may or may not have an abundance of various lysis-related proteins.

There are 645/2035 cells with identified pRE foci. There are 141/2035 cells with both pRE and pR' foci, so 504/2035 cells with pRE without pR' foci. Note that these cells were grown in LB. We have updated Fig. 5 with these numbers (In 783-785). As a comparison, we counted 45 lysogens in our movies out of 204 cells grown in M9 under the microscope.

As for the proposed question, that is a question that we cannot fully answer with just these data. Our voting model was based on live-cell data where cells with gpD expression were precluded from establishing lysogeny. There is not a quantitative link between the level of pR' signal during FISH and the presence of our live-cell gpD reporter. To speculate, we would expect that very low levels of pR' activity can be overcome if there is enough CI activity, but we know that it is rare for lysis to be halted if the gpD reporter is detected (Trinh, JT, et al. Nat Commun. 2017).

19. Line 233-234: 'we showed that phage DNAs organize their transcripts locally' please complete the sentence with 'at early time points', or 'at 15 min'.

Response: We have changed this (In 246).

20. Suppl. text, page 2, 6 lines from bottom, 'for example in Fig1e, 70 min', there is no 70 min time point in this figure. Is it 80min?

Response: Yes, it was meant to be 80 min, and we have changed this (Supp. In 60).

21. Suppl. Fig11, title of the legend: 'DNA and which', remove 'and'

Response: We have changed this (Supp. In 471).

22. Suppl. Fig11, legend b-d. The lytic cells' data are probably from Fig. 2h, rather than 3h.

Response: We have fixed it (Supp. In 476).

Reviewer #3

We would like to express our appreciation for Reviewer #3's careful reading of the manuscript. We are very happy that this reviewer considered our work to be "of good quality" and to offer an "interesting new insight" into stochastic processes in bacteria. The reviewer had a few questions/concerns which we will do our best to address here.

1. Could the authors elaborate why their DnaB fusion does not form foci in normal growing cells as would be expected from ongoing chromosomal replication in these conditions? I would imagine replication foci should be visible given the high-quality microscope and EMCCD camera used in this study: for example, a recent paper on *E. coli* replication dynamics after UV damage a YPet-DnaB forms clear foci under normal growth conditions (Soubry N, Wang A, and Reyes-Lamothe R (2019) Replisome activity slowdown after exposure to ultraviolet light in *Escherichia coli*. PNAS 116(24):11747-11753).

Response: We thank the reviewer for highlighting the recent work in the field of single-cell dynamics. Soubry, N, et al. PNAS. 2019 used a laser light source and a TIRF setup, which is known to have less fluorescence background, unlike our widefield epifluorescent setup. Their methodology is established to be specialized for the detection of very low-copy molecules, unlike ours. In addition, they used a yellow fluorescent protein, YPet, which is brighter than cyan-colored fluorescent proteins, such as mTurquoise2, which we used. Under our imaging conditions, we can see clear foci when we have at least 40 mTurquoise2 FPs, so we were not able to detect a single hexamer of DnaB.

2. Was this fusion verified by sequencing, and does this fusion have the same growth rate and average cell size to the WT?

Response: The fusion was verified by PCR and microscopy. We tested for insertion using several sets of primers. The PCR gels suggest a correct insertion at the recombination site. Details for this have been added to Supplementary Text (Supp. In 257-275) and Supp. Fig. 2f. The PCR product also has the correct sequence. The labeled cells show a distinct phenotype during imaging, and we were able to transduce this construct into other strains and recapitulate this phenotype.

Cells with the *dnaB-FP* allele have similar growth rate to MG1655, with a 26 min doubling time grown in LB, suggesting no major ill effects from the DnaB-FP fusion. Cells with the *dnaB-FP* allele are similar to (slightly larger than) an isogenic strain. We have included the details of this experiment in Supplementary Text (In 276-296) and Supp. Fig. 2g, j, and referred to in the text (In 69-70).

3. If possible, I would also suggest flow cytometry to verify the number of chromosomes per cell is the same as WT. Can they rule out having selected for a gene duplication with an unlabeled DnaB copy during the red-recombination?

Response: Our cells grew at a normal rate, as noted above (Supp. Fig. 2g and j). We also note that the *dnaB-FP* strain was relatively simple to construct, with good efficiency during red recombination and it transduced easily. Our new growth data suggest that the DnaB-FP construct is not detrimental, such that it would not pressure the cell to duplicate an essential gene. To note, when we were constructing the DnaB-FP reporter, the *dnaB-FP* gene was put on a high-copy number plasmid with leaky expression, and there was no significant effect on cell growth, which also suggests that the reporter does not impose strong negative effects. Our PCR results suggest that the *dnaB-mTurquoise2* allele is correctly inserted, and that a second copy of *dnaB* is not adjacent (Supp. Fig. 2f). In addition, we have performed DNA FISH experiments against near *attB* region. The DnaB-mTurquoise2 fusion strain (LZ1511) has a similar average

number of attB sites (1.96 ± 0.92) as WT, MG1655 (1.97 ± 0.83). From our data, we believe it is highly unlikely for a duplication event to have occurred.

4. page 5 line 95: 'time points' is not a very useful measure – please use minutes.

Response: We have changed this (In 97).

5. The authors conclude that the *E. coli* chromosome provides a barrier which determines phactory location. However, in the later FISH experiments they use rifampicin treatment and find that phactory organization appears to remain. Rifampicin causes a rapid loss of nucleoid organization and chromosomal DNA fills the whole cell volume after treatment (See Cabrera, J. E., & Jin, D. J. (2003). The distribution of RNA polymerase in *Escherichia coli* is dynamic and sensitive to environmental cues. *Molecular Microbiology*, 50(5), 1493–1505). If the nucleoid was the primary barrier in Wephactory formation I would have hypothesized that rif treatment would lead to a loss of phactory organization. From the images in Supplementary Fig. 15a-e, it appears that the nucleoid structure is retained after rif treatment used, so perhaps the concentration is not high enough to see this effect. In the attB experiments I would suggest incubating with a higher concentration of rif which does causes nucleoid structure loss (this can be verified by DAPI), to see if bacterial and phage DNA separation is lost. This would help verify their hypothesis.

Response: This is a correct point, that nucleoid organization is characterized to change following rifampicin treatment. That study (Cabrera, JE and Jin, DJ. *Mol Microbiol.* 2003) observed *E. coli* nucleoid expansion after 10 min with 50 µg/mL rifampicin grown in LB at 30°C. In a later study from the same group (Cabrera, JE, et al. *J Bacteriol.* 2009), they observed that *E. coli* nucleoid expansion plateaus (90% relative size) after 20 min with 100 µg/mL rifampicin, grown in LB at 32°C, and 70% relative size after 10 min of that treatment, but the *E. coli* nucleoid does not fill the whole cell. This suggests that nucleoid expansion is a dynamic process. Another group (Bakshi, S, et al. *Mol Microbiol.* 2014) reported that *E. coli* nucleoid expansion due to rifampicin treatment is a function of time. Therefore, we think that *E. coli* expansion is not complete in our experiments following 15 min treatment at 50 µg/mL rifampicin.

To test this, we examined the *E. coli* nucleoid morphology by treating the cells with 50 µg/mL, 100 µg/mL and 300 µg/mL rifampicin for 0, 15, and 30 min. At 50 µg/mL, the *E. coli* nucleoid still shows an intermediate level of organization at 15 min, but increasingly fills the whole cell at 30 min. Higher concentrations of rifampicin induce more expansion by 15 min. This result has been included in Supp. Fig. 15l and described in the Methods (In 429-440). The results confirm that our rifampicin treatment is properly inhibiting bacterial and phage transcription because the expected downstream influence on the nucleoid is intact. Thus, the results from Supp. Fig. 15a-k followed our experimental design and support our conclusion.

Furthermore, we performed another infection RNA FISH experiment using 300 µg/mL rifampicin. We compared the cells at 0 min and 15 min after addition of rifampicin, which induces nucleoid expansion in the absence of phage. We observed that at 15 min, there was still organization of the phage RNAs and that DAPI was still mostly organized subcellularly. The population is perhaps intermediately organized compared to 0 min after rifampicin, but organization persists, nonetheless. This data was added to Supp. Fig. 15m. We believe this is a reasonable outcome and can explain the results in the context of our model.

We want to make an important point here regarding the reviewer's hypothesis "that rif treatment would lead to a loss of phactory organization." This would only be true under the specific

assumption that active transcription is needed for the phage DNA/RNA to remain organized, as it does for bacterial DNA. We do not believe there is evidence for this assumption.

To elaborate, we claim that the phage phactory is essentially not miscible with *E. coli* DNA. We also observe that the phage DNA expands in space and generates a force pushing against the bacterial DNA. This is important, as active transcription does not appear to have a condensing effect on phage DNA, given the expansion of it. The amount of phage DNA in the phactory is similar to the amount of DNA in the bacterial chromosome. Lambda has a 48.5 Kbp genome compared to the 4.6 Mbp genome of *E. coli*. The burst size of lambda is ~100 phages, so in relative terms, phage and bacterial DNA levels are similar. However, phage DNA takes up more space than bacterial DNA. This suggests that the mechanisms which compact *E. coli* DNA are not active on phage DNA, and that the phage-induced changes in *E. coli* nucleoid organization may be maintained in the presence of rifampicin. In other words, nucleoid reorganization from rifampicin may not fully override nucleoid reorganization by the phage activity.

6. The distinct gene expression profile between phactories in the same cell is an important aspect of this work, but the authors need to do more to quantitatively show that the compartmentalization they describe is responsible for this. In Fig. 4, panel f,g, representative cells are shown from 48/2035 cells where different compartments have exclusively pR' or pRE signal. What percentage cells have both signals in the same compartment? For cells which have both signals are they significantly more likely to be located in different compartments? How does the heterogeneity in expression between same cell compartments compare to between cells? Are more spatial distant compartments more likely to make different decisions?

Response: There are 141/2035 cells with both pRE and pR' foci, so 93 of those cells show both signals in same compartments. Roughly two-thirds of mixed pRE and pR' cells have the foci in same cell locations and one-third of the cells have them separated, so it is more likely that the different transcripts will overlap, at this time point. We added these data in Fig. 5 (ln 787-788).

Regarding within-cell vs between-cell heterogeneity, we performed an analysis where we compared the pR signals from one cell half versus the other. Variability in pR is generally thought to influence downstream decision-making at the cell-level. This analysis produces a distribution of within-cell ratios. We compared this within-cell variation to between-cell variation by bootstrapping pairs of cells from our dataset, essentially attempting to reproduce our within-cell analysis with whole-cell signals. Via bootstrapping, we obtained a median of the medians of between-cell variations to compare with our within-cell data. Briefly, within-cell pR levels vary more than between cells. We believe this analysis properly captures the data because when we did this analysis with DAPI signals, within-cell variation was less than between-cell variation. DAPI staining is expected to vary cell-to-cell, where some cells take up less dye, but a single cell would not be expected to have more DAPI in one half of the cell. Regarding pR, this analysis suggests that there is a bias towards pR RNA being at different levels on one side of the cell versus the other. These data have been included in Supp. Fig. 14g and in the main text (ln 215-217), and the analysis was explained in the Supplementary Text (Supp. ln 223-246).

7. The authors could try using subinhibitory concentration of cephalexin prior to infection to generate filamentous cells containing many separate nucleoids (for examples see Nonejuie et al (2013). Bacterial cytological profiling rapidly identifies the cellular pathways targeted by antibacterial molecules. PNAS, 110(40), 16169–16174). These should be enriched for single cells making multiple decisions.

Response: In this work, our goal was to establish this phactory model, and while the proposed experiment would be very interesting and support our model, we feel such experiments would be beyond the scope of this work. It will be interesting to examine this in the future. We feel that our data are sufficient to establish that different centers of phage activity emerge in cells of normal size. Probing larger cells feels like an extension and could be combined with other experiments for a separate study.

8. In general, the retention of phage RNA near its site of origin even after rifampicin treatment is very interesting - do the authors speculate that this behavior is unique to phage mRNA, or would host mRNA be similarly confined within the same 'compartments'? If this is simply caused by entropic forces preventing 70S ribosomes diffusing through the nucleoid (as seen by Sanamrad et al (2014). Single-particle tracking reveals that free ribosomal subunits are not excluded from the *Escherichia coli* nucleoid. PNAS) then non-phage mRNA would also be expected to be similarly compartmentalized. This would be interesting to look at in future work.

Response: This observation of phage mRNA organization was unexpected and was made long before we thought about this concerted model of phage DNA, RNA, and virions in subcellular compartments. For the spatial aspect of lambda, the phages are reported to prefer the cell poles and mid-cells during the adsorption step before infection (Edgar, R, et al. Mol Microbiol. 2008). We found that our phage DNAs also prefer the cell poles and mid-cell during the initial time periods of infection, and that the phage RNAs are localized similarly. For lambda, some transcripts are quite long, and will remain tethered to its cognate DNA for minutes under maximum reported transcription rates (Vogel, U and Jensen, KF. J Bacteriol. 1994). These locations are enriched with ribosomes in *E. coli* (Bakshi, S, et al. Mol Microbiol. 2012), so it stands to reason that translation will simultaneously occur for the phages here as well. In other words, there is a lot of phage-specific mass including DNA, RNA, and protein, which might help the phactory stick to itself, be hindered in translocating, and separate from the host nucleoids. We agree that it will be interesting to examine non-phage mRNA in the future.

REVIEWERS' COMMENTS:

Reviewer #2 (Remarks to the Author):

The questions have been answered, th emanuscript is now improved and can be accepeted, from my point of view.

Reviewer #3 (Remarks to the Author):

The authors have done a good job responding to the comments. The new comparison of within-cell variation to between-cell variation of pRE and pR' foci was an important control to rule out stochastic staining / imaging confounding their results. Similarly the more thorough deconvolution of the effects of rifampicin treatment (i.e. halting transcription and perturbing nucleoid organization) strengthens the work.

Detailed Responses to Reviewers' Comments

Reviewer #2 (Remarks to the Author): The questions have been answered, the manuscript is now improved and can be accepted, from my point of view.

Response: We are happy to see that the reviewer is satisfied with our changes.

Reviewer #3 (Remarks to the Author): The authors have done a good job responding to the comments. The new comparison of within-cell variation to between-cell variation of pRE and pR' foci was an important control to rule out stochastic staining / imaging confounding their results. Similarly the more thorough deconvolution of the effects of rifampicin treatment (i.e. halting transcription and perturbing nucleoid organization) strengthens the work.

Response: We are happy to see that the reviewer is satisfied with our changes.

Responses to the additional reviews from Reviewer #3

We would like to extend our sincerest gratitude towards Reviewer #3 for his or her additional efforts in reviewing Reviewer #1's review. We are pleased with the concordant opinions from Reviewer #3 in support of our experimental design and rationale being proper and satisfactory for our work.

1. The manuscript is hard to read as it is not subdivided in sections with subtitles that would make the storyline and arguments more convincing. **Response: We have divided our writing into clear subsections. Thanks for the suggestion.**

I think the revised manuscript is easier to read.

Response: We appreciate the positive comment.

2. The supposed individuality they see could be simply a result of looking at cells that are not synchronized in terms of cell cycle. **Response: We agree that different cells may be at different stages at a given time point in our images despite our efforts to synchronize them as described in the Methods. We do not believe that this undercuts any of our conclusions. Our model is phagedriven, and the phage will develop regardless of the status of the cell cycle. We showed that the phages develop in individual compartments across different cells, which encompass different states of the cell cycle. Therefore, our claims of individuality are independent of the cell cycle.**

I would imagine that the stage in the cell cycle of a given cell will determine the position and number of available 'compartments' in which the phage can initially develop. E.g. in minimal media newborn cells will have one central nucleoid, therefore separating phage into 2 possible initial compartments, late C period cells will have 2 nucleoids so three possible compartments. The number of cells in each stage will in turn probably impact the fraction of 'push' vs 'squeeze' vs 'spread' in Fig. 3. Since the heterogeneity the authors are referring to is between the

compartments within a single cell this does not detract from their conclusions, since even the newborn cells should have 2 possible compartments.

Response: We appreciate the understanding of our work.

3. It is not clear whether the fluorescent labels they use do not cause aggregation and thus artefacts. There is a famous paper from Johan Paulsson lab at Harvard which attest to such issues of aggregates. (Landgraf et al Nat Methods 2012). **Response: This is a cogent point, as spurious aggregation of fluorescent proteins could confound our conclusions. We do not believe this occurs in our work. Our constructs have either been validated previously (e.g. TetR reporter: Lau, IF, et al. Mol Microbiol. 2003; Wang, X, et al. Genes Dev. 2005. SeqA reporter: Babic, A, et al. Science. 2008; Shao, Q, et al. Biophys J. 2015; Shao, Q, et al. MicrobiologyOpen. 2016; Trinh, JT, et al. Nat Commun. 2017. gpD reporter: Zeng, L, et al. Cell 2010; Trinh, JT, et al. Nat Commun. 2017) or we have validated in this work through extensive control experiments (e.g. DnaB construct, the new Supp. Fig. 2). We have also included FISH experiments which do not involve fluorescent proteins to support our data in the manuscript.**

Since the DnaB, TetR and GpD fusions do not form foci in the absence of phage (which is now clearer in the text), this provides reasonable evidence that they are not prone to aggregation. It is very difficult to completely prove there is no aggregation effect for any given fusion, aside from testing a wide variety of different constructs / fusions as in the Landgraf et al paper mentioned by reviewer 1, which is realistically too burdensome to request of the authors. But since aggregation artifacts are a well-known potential problem with any work involving fluorescent fusions proteins, most readers will be aware of this limitation.

Response: We appreciate the understanding of our work.

4. The interpretation of individuality has no consequence on the nature of individual phages as they are all the same after all, so I do not see any biological relevance. Many things are stochastic in cells yet not of biological relevance. **Response: We respectfully disagree that individuality has yet not of biological relevance. It made a huge difference whether phages act as a mass-action pool or phages as individual decision makers, which lead to gene dosage mechanism or individual phage voting mechanism respectively. These differences were published in Zeng, L, et al, Cell 2010. In our previous work (Trinh, JT, et al. Nat Commun. 2017), we found interesting phage-phage interactions inside the cell including competition and cooperation which is another evidence of the interesting behaviors by individual phages, and in this work we elaborate the inner workings of how individual phages behave at the subcellular level in depth.**

I disagree with reviewer 1's assessment. The same argument could be made about the isogenic population of *E. coli* that the phages are infecting, yet there is a wealth of excellent research highlighting biologically relevant stochastic behavior and cell to cell heterogeneity in the literature.

Response: We appreciate the support.

5. Furthermore, in order to see how much the nucleoid itself modulates what the authors see, I would suggest looking at minimal media grown cells versus rich media grown cells, where nucleoid dynamics and compaction vary dramatically and to decorate the nucleoid fully (eg FisGFP John Marko lab has published work that can be of help for such experiments, Yazdi et al Mol Microbiol 2012) such that we can clearly see what spaces are free. **Response: Indeed, different growth conditions can affect the physiology of the cell and the phage consequently. Our live-cell time-lapse infection experiments (Fig. 1, 3) were done using minimal media, our live-cell non-time lapse induction experiments (Fig. 2) and our fixed-cell imaging experiments (DNA FISH, Fig. 4; RNA FISH, Fig. 5) were done using rich media. We have varied our growth conditions and these experiments converge to the same conclusions. This is a reasonable rebuttal.**

Response: We appreciate the support.

6. I also have a hard time with the many anthropomorphic terms. A factory relies on input of energy, planning, and much more than self-assembly. To talk about “communities” of self-assembling molecules I also find misleading. Just by introducing new terminology to effects that have already an old nomenclature does not make for new science. I know that there is a trend of finding new catch words, but this has really been abused. **Response: We respect the philosophical differences on how to describe biological processes. Specifically, regarding the comment on a phage “factory,” this is not a new term or way of thinking about phage biological processes. Long ago, researchers have questioned in different phage, the topology of phage gene products and hypothesized that localization of these enzymes could resemble a subcellular factory (Kozinski, AW and Kozinski, PB, PNAS. 1967). In this work, our data reflect restrictive position, co-localization, and recruitment of host resource for phage development, etc, so we feel that a phage factory can be used here. We termed as “phactory” using “ph” uniquely for phage. On the comment about “communities,” we agree that perhaps this invokes more complexity than intended in this manuscript. We decided to use “microenvironment” instead, which may be a better fit in this work. This is a reasonable rebuttal.**

Response: We appreciate the support.

7. Throughout the manuscript it is never clear when we look at single infections or multiple infections, thus some basic aspects are not clear. **Response: The average MOI is ~4 for the infection experiments (added to the Methods). The individual MOI of each cell varies according to the Poisson distribution of phage adsorption to the cell surface. In this manuscript, we did not specifically determine the MOI of each cell since the driver of individuality is the individual phage DNA, not the infecting viral particle on the cell. Due to the high rate of failed infection (20-30% published in Zeng, L et al, Cell. 2010; Trinh, JT, et al, Nat Commun. 2017), MOI is not an accurate count of successfully injected DNA. In addition, the individual phage DNA inside the cell can be the injected phage DNA and/or replicated copies of the injected DNA. In other words, the conclusions for this study do not depend on the number of infections, only on the**

number and location of DNAs inside the cell at the early time point upon infection. We also drew the same conclusions from cells that were not infected at all, which underscores this point, using lysogenic induction to show the localized phage development (new Fig. 2).

This is a reasonable rebuttal.

Response: We appreciate the support.

8. Also throughout it is completely not clear what is cause and what effect, what we see is due to bacterial biology or to some intrinsic phage mechanism? Is it happenstance or is there a mechanism that has some consequence/advantage that it confers to the virus on a global level or individual one. What are the implications of individuality if any? **Response: We think it is a combination of bacterial biology and phage mechanisms. For example, we stated that the phage alters the natural behavior of DnaB, the phage maintains its clones and resources, the expansion of phage DNA was a phage-active mechanism, and the phages execute divergent developmental pathways. Phages are viruses, so they neither fully de-couple from, nor fully rely on the host. We believe this work represents a point from which multiple projects can branch from, rooted in these findings that even identical phages can act as individuals in a cell. As is, the results are consequential for providing mechanisms for earlier questions and establishing a new perspective to think about this, and other systems moving forward.**

This is a reasonable rebuttal.

Response: We appreciate the support.